# Solid Angle Geometry-Based Modeling of Volume Scattering with Application in the Adaptive Decomposition of GF-3 Data of Sea Ice in Antarctica

**Dong Li** [1,2] , **He Lu** [1,2] **and Yunhua Zhang** [1,2,*]

1   CAS Key Laboratory of Microwave Remote Sensing, National Space Science Center,
    Chinese Academy of Sciences, Beijing 100190, China; lidong@mirslab.cn (D.L.);
    luhe20@mails.ucas.ac.cn (H.L.)
2   School of Electronic, Electrical and Communication Engineering, University of Chinese Academy of Sciences,
    Beijing 100049, China
*   Correspondence: zhangyunhua@mirslab.cn

**Abstract:** Over the last two decades, spaceborne polarimetric synthetic aperture radar (PolSAR) has been widely used to penetrate sea ice surfaces to achieve fully polarimetric high-resolution imaging at all times of day and in a range of weather conditions. Model-based polarimetric decomposition is a powerful tool used to extract useful physical and geometric information about sea ice from the matrix datasets acquired by PolSAR. The volume scattering of sea ice is usually modeled as the incoherent average of scatterings of a large volume of oriented ellipsoid particles that are uniformly distributed in 3D space. This uniform spatial distribution is often approximated as a uniform orientation distribution (UOD), i.e., the particles are uniformly oriented in all directions. This is achieved in the existing literature by ensuring the canting angle $\varphi$ and tilt angle $\tau$ of particles uniformly distributed in their respective ranges and introducing a factor $\cos \tau$ in the ensemble average. However, we find this implementation of UOD is not always effective, while a real UOD can be realized by distributing the solid angles of particles uniformly in 3D space. By deriving the total solid angle of the canting-tilt cell spanned by particles and combining the differential relationship between solid angle and Euler angles $\varphi$ and $\tau$, a complete expression of the joint probability density function $p(\varphi, \tau)$ that can always ensure the uniform orientation of particles of sea ice is realized. By ensemble integrating the coherency matrix of $(\varphi, \tau)$-oriented particle with $p(\varphi, \tau)$, a generalized modeling of the volume coherency matrix of 3D uniformly oriented spheroid particles is obtained, which covers factors such as radar observation geometry, particle shape, canting geometry, tilt geometry and transmission effect in a multiplicative way. The existing volume scattering models of sea ice constitute special cases. The performance of the model in the characterization of the volume behaviors was investigated via simulations on a volume of oblate and prolate particles with the differential reflectivity $Z_{DR}$, polarimetric entropy $H$ and scattering $\alpha$ angle as descriptors. Based on the model, several interesting orientation geometries were also studied, including the aligned orientation, complement tilt geometry and reflection symmetry, among which the complement tilt geometry is specifically highlighted. It involves three volume models that correspond to the horizontal tilt, vertical tilt and random tilt of particles within sea ice, respectively. To match the models to PolSAR data for adaptive decomposition, two selection strategies are provided. One is based on $Z_{DR}$, and the other is based on the maximum power fitting. The scattering power that reduces the rank of coherency matrix by exactly one without violating the physical realizability condition is obtained to make full use of the polarimetric scattering information. Both the models and decomposition were finally validated on the Gaofen-3 PolSAR data of a young ice area in Prydz Bay, Antarctica. The adaptive decomposition result demonstrates not only the dominant vertical tilt preference of brine inclusions within sea ice, but also the subordinate random tilt preference and non-negligible horizontal tilt preference, which are consistent with the geometric selection mechanism that the c-axes of polycrystallines within sea ice would gradually align with depth. The experiment also indicates that, compared to the strategy based on $Z_{DR}$, the maximum power fitting is preferable because it is entirely driven by the model and data and is independent of any empirical thresholds. Such soft thresholding enables this strategy to adaptively estimate the

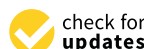



negative $Z_{DR}$ offset introduced by the transmission effect, which provides a novel inversion of the refractive index of sea ice based on polarimetric model-based decomposition.

**Keywords:** polarimetric decomposition; radar polarimetry; synthetic aperture radar; sea ice; volume scattering model

## 1. Introduction

Sea ice covers approximately $2.5 \times 10^7$ km$^2$ of the Earth's surface and accounts for 5~8% of the global ocean area. They are mainly distributed in the high latitudes of the North and South Poles, such as the Sea of Okhotsk and the marginal sea area of the Antarctic. Sea ice makes the Arctic and Antarctic cold sources, which impact the exchange of energy between the ocean and the atmosphere and affect global water circulation and climate change [1]. Moreover, the unfreezing, refreezing, and drifting of sea ice will be destructive to human activities and cause serious losses for the marine economy [2,3]. Therefore, the efficient and accurate monitoring of sea ice is critical.

The wide distribution of sea ice and the harsh environments at the Poles make in situ observation of sea ice all but impossible [4]. As a result, remote sensing techniques have been introduced [5,6], Microwave remote sensing has attracted particular interest due to its all-day and all-weather capabilities [7–11]. Synthetic aperture radar (SAR), as an active microwave remote sensing technique, can penetrate sea ice surfaces for high-resolution microwave imaging [12]. Gaofen-3 (GF-3), launched in 2016 is China's first high-resolution SAR satellite at C-band, which can acquire the polarimetric SAR (PolSAR) image of sea ice by alternatively transmitting and simultaneously receiving orthogonal polarized electromagnetic waves (EMW). This makes the polarimetric decomposition of sea ice possible, since the polarization of EMW is sensitive to the dielectric properties and geometric structure of sea ice. Although GF-3 data have been used for sea ice detection and classification [13,14], there has been almost no research using GF-3 PolSAR data for polar sea ice scattering decomposition so far.

The polarimetric scattering information of sea ice acquired via PolSAR is often a $3 \times 3$ covariance matrix or coherency matrix, which cannot be directly applied for identification and classification unless we resort to specific matrix analysis approaches. Polarimetric target incoherent decomposition is just such an approach. It is dedicated to pursuing the geophysical scattering mechanism of an unknown scatterer by extracting a dominant single target (such as the phenomenological dichotomies [15–18] and the eigenvector-based decompositions [19–21]) or expanding the scatterer on specific canonical models (such as model-based decompositions [22–39]). Among these approaches, the model-based decompositions can be efficiently implemented with clear physical significance. They have previously been used for the classification and inversion of depth of sea ice, separating different scattering components from the backscattering echoes [40–51]. Zhang et al. [42] used a double-bounce scattering component to effectively distinguish gray ice from fast ice. Shokr et al. [50] analyzed the scattering mechanisms of different kinds of sea ice and used the ratio of volume scattering power to surface scattering power to distinguish rough first year ice from smooth first-year ice. He et al. [51] extracted the polarimetric features using four typical model-based decompositions and input them into the random forest algorithm to classify open water and sea ice. Zhang et al. [45] used the volume scattering power parameters to invert the depth of sea ice in the Bohai Sea. Shokr and Dabboor [47] estimated the thickness of fast ice using four-component decomposition parameters. Parrella et al. [48] analyzed the microstructure of glacial ice anisotropy based on the constructed sea ice volume scattering model, and used the co-polarization phase differences to invert the firn depth.

Volume scattering is an indispensable component of model-based decomposition. Sea ice is a polycrystalline medium, in which each polycrystal is composed of many small

ice platelets [52]. The seawater trapped in pockets between these ice platelets become brine inclusions as the ice grows and the temperature decreases [52]. Each brine inclusion has a substantially ellipsoidal shape and a high permittivity [53]. The polarimetric volume scattering of sea ice, especially that of the young ice, is mainly induced by these differently oriented brine inclusions. Plenty of models have been developed for natural terrains [22,23,54–57], but for now, we will put these aside. Regarding the diversification of scattering modeling for sea ice, in 1995, Rignot et al. [58] suggested modeling the scattering of Greenland ice sheet as randomly oriented cylinders embedded in transparent snow medium. However, poor agreement between PolSAR observation and model was found at the L- and P-band [59]. Sharma et al. [40] in 2010 stressed that the 2D orientation around the radar line of sight (LOS) described by Rignot et al. [58], Freeman and Durden [22] and Yamaguchi et al. [23] is insufficient to model the 3D orientation geometry of ice particles in the Earth-based Cartesian coordinate system. Hence, they introduced two Euler angles, i.e., the canting angle $\varphi$ and the tilt angle $\tau$, to describe the 3D orientation of ice particles. The volume covariance/coherency matrix is estimated by ensemble averaging the covariance/coherency matrix of $(\varphi, \tau)$-oriented particle under the joint probability density function (PDF) $p(\varphi, \tau)$ of Euler angles $\varphi$ and $\tau$. A similar volume scattering modeling technique was used by Zhang et al. [45] in 2014 to analyze the polarimetric scattering of ice in the Bohai Sea, China. Nevertheless, the modeling of Sharma et al. did not adequately explain the co-polarization phase difference often observed in PolSAR datasets because Sharma et al. modeled the sea ice particle as a simple dipole [40]. To improve this, based on the previous work of Cloude et al. [60], Parrella et al. (2015) [46,59] treated ice inclusions as ellipsoid particles for volume modeling. In a more recent work, Parrella et al. [49] validated that this model allows characterization of the main scattering mechanism in different glacier zones and provides a clear link to the respective subsurface structures.

This article revisits the volume scattering modeling of sea ice from the perspective of statistical distribution of particle orientation, i.e., $p(\varphi, \tau)$, which is related to the 3D spatial distribution $p(r, \varphi, \tau)$ of particles. The contribution of distance $r$ between radar and particles is mainly reflected in the extinction effect, which is independent of the orientation $(\varphi, \tau)$-induced volume scattering modeling discussed in this paper. The sea ice particles are usually assumed to follow a uniform spatial distribution (USD) [40,46,49,59]. In view of the independence between orientation and distance, the USD of $p(r, \varphi, \tau)$ also implies the uniform orientation distribution (UOD) of $p(\varphi, \tau)$, i.e., the sea ice particles are uniformly oriented in all directions. Sharma et al. [40] provided an implementation of UOD for scattering modeling of sea ice by ensuring the canting angle $\varphi$ and tilt angle $\tau$ of particles uniformly distributed in their respective ranges and including a factor $\cos \tau$ in the ensemble average. This implementation has also been used in a series of works [46,49] by Parrella et al. Nonetheless, we find this implementation of UOD is not always effective; a real UOD can be only realized by distributing the solid angles of particles uniformly in 3D space. By deriving the total solid angle of the canting-tilt cell spanned by particles and combining the differential relationship between solid angle and Euler angles, a joint PDF $p(\varphi, \tau)$ for UOD is realized, which can not only ensure the uniform orientation of particles, but also completely covers all situations of orientation distributions of particles of sea ice. By ensemble integrating the obtained $p(\varphi, \tau)$ into the coherency matrix of $(\varphi, \tau)$-oriented ellipsoid particle established on the small particle scattering and transformation among radar polarization coordinate system, ellipsoid coordinate system and Earth-based Cartesian coordinate system, a generalized modeling of the volume coherency matrix of a cloud of 3D uniformly oriented spheroid particles is then attained. The model covers factors such as radar imaging geometry, particle shape, particle canting geometry, tilt geometry and transmission effect in a multiplicative way. It describes the polarimetric volume scattering of the typical distributed targets such as vegetation covers, soil particles and ice inclusions. Meanwhile, the existing volume models of sea ice are only applicable to specific instances. The successful performance of the volume model is validated by

simulations on a cloud of oblate and prolate particles using the differential reflectivity $Z_{DR}$, polarimetric entropy $H$ and scattering $\alpha$ angle as descriptors.

One important application of scattering modeling is to better describe the underlying components of sea ice scattering, while the related parameters such as scattering powers are achieved by the polarimetric model-based decomposition. Sharma et al. [40] proposed decomposing the scatterings of glacier ice into the incoherent sum of the surface, sastrugi and volume components. A scattering balance equation system with thirteen unknowns was constructed and solved numerically with a series of approximations and simplification to match the nine degrees of freedoms (DoF) in the covariance/coherency matrix. Nevertheless, there is an additive residual component in the decomposition which can be minimized to the $L_2$ norm but cannot be eliminated. A similar residue also exists in the decomposition of Parrella et al. [46], where the backscattering of glacier and ice sheets is simplified as the incoherent addition of the X-Bragg surface scattering and volume scattering. This kind of incomplete utilization of polarimetric DoF is often coupled with the problem of negative scattering power in other decompositions, such as Freeman-Durden three-component decomposition (FDD). To overcome the negative power, in 2011, van Zyl et al. [28] devised the nonnegative eigenvalue decomposition (NNED), which was extended to the scattering of non-reflection symmetry (NS) by Liu et al. [33] and Wang et al. [34]. Zhang et al. [45] used NNED-NS in the model-based interpretation of ice scattering by additively decomposing the backscattering into the surface, double-bounce and volume components. A remainder, however, is still inevitable despite the nonnegative eigenvalues. In addition to NNED-NS, the complete model-based decomposition (CMD) designed by Cui et al. [31] in 2013 is also an extension of NNED for NS. However, unlike NNED-NS, CMD not only solves the problem of negative power in model-based decomposition but also results in full use of scattering DoFs. To demonstrate the application of the proposed models in the volume decomposition of the sea ice scattering, following CMD, the scattering power that reduces the rank of coherency matrix by one without breaking the physical realizability is obtained to make full use of the polarimetric information. Moreover, to facilitate the calculations, the existing sea ice decompositions only consider a model of fixed orientation, such as the randomly oriented model described by Sharma et al. [40] or the vertically tilted model discussed in the works of Zhang et al. [45] and Parrella et al. [46]. Vertical, random and horizontal orientations are all possible for sea ice particles. To achieve this, based on the generalized model, some interesting orientation geometries are also studied, including the aligned orientation, complement tilt geometry, reflection symmetry and azimuth symmetry. The complement tilt geometry is of particular interest, as it provides three models that correspond to the horizontal tilt, vertical tilt and random tilt of sea ice particles, respectively. To match the models to PolSAR data for adaptive decomposition, two strategies are provided. One is based on $Z_{DR}$ and the other is based on the maximum power fitting. Both the strategies and models are validated on the GF-3 PolSAR dataset of a young ice area in Prydz Bay, Antarctica. The adaptive decomposition result displays not only the dominant vertical tilt preference of sea ice brine inclusions, but also the subordinate random tilt preference and non-negligible horizontal tilt preference, which are consistent with the geometric selection mechanism in that the c-axes of polycrystalline formations within sea ice gradually align with the depth. The experiment also shows that compared to the $Z_{DR}$-based strategy, the maximum power fitting is preferable, as it is fully driven by model and data, and independent of any empirical threshold. Such soft thresholding enables it to adaptively retrieve the $Z_{DR}$ offset introduced by the transmission effect, which provides a new inversion of the refractive index of sea ice based on polarimetric model-based decomposition.

The remainder of this article is arranged as follows. Starting from the theory of small particle scattering, the coherent scattering of a $(\varphi, \tau)$-oriented ellipsoid particle is modeled in Section 2 based on the transformation among the radar polarization coordinate system, ellipsoid coordinate system and Earth-based Cartesian coordinate system. Section 3 determines the joint PDF $p(\varphi, \tau)$ of Euler angles $\varphi$ and $\tau$ for UOD based on the solid angle geometry. The multiplicative generalized modeling of the coherency matrix of a volume

of 3D uniformly oriented spheroid particles is performed and simulated in Section 4 by ensemble integrating $p(\varphi, \tau)$ with the coherency matrix of the $(\varphi, \tau)$-oriented ellipsoid particle. To exemplify the application of the proposed models, Section 5 presents the adaptive polarimetric decompositions of the volume scattering component for sea ice. Finally, both the adaptive decompositions and models are validated in Section 6 on the GF-3 PolSAR data of a young ice area in Prydz Bay, Antarctica. Section 7 concludes the article. The matrices $[A]_{mn}$ and $[B]_{mn}$ that comprise the essential matrix $[\omega]_{mn}$ are formulated in Appendix A.

## 2. Coherent Scattering Modeling for a 3D Oriented Ellipsoidal Particle

The volume scattering of sea ice is generally considered to be induced by the complex brine inclusions within the subsurface layer [40,45], as shown in Figure 1. The modeling of such volume effects is usually challenging due to the complicated interaction of EMW and the complex composition and distribution of brine inclusions. To simplify the volume modeling, a commonly-used solution is to treat each localized brine scattering center as an ellipsoidal particle of identical shape, size and composition embedded in the homogenous background [40,45]. The problem is then reduced to establishing the polarimetric coherent scattering of a particle and combining a large volume of such independent elements to obtain a second-order statistical description of the scattering medium. We begin by focusing on the coherent scattering modeling of a 3D-oriented ellipsoid in this section.

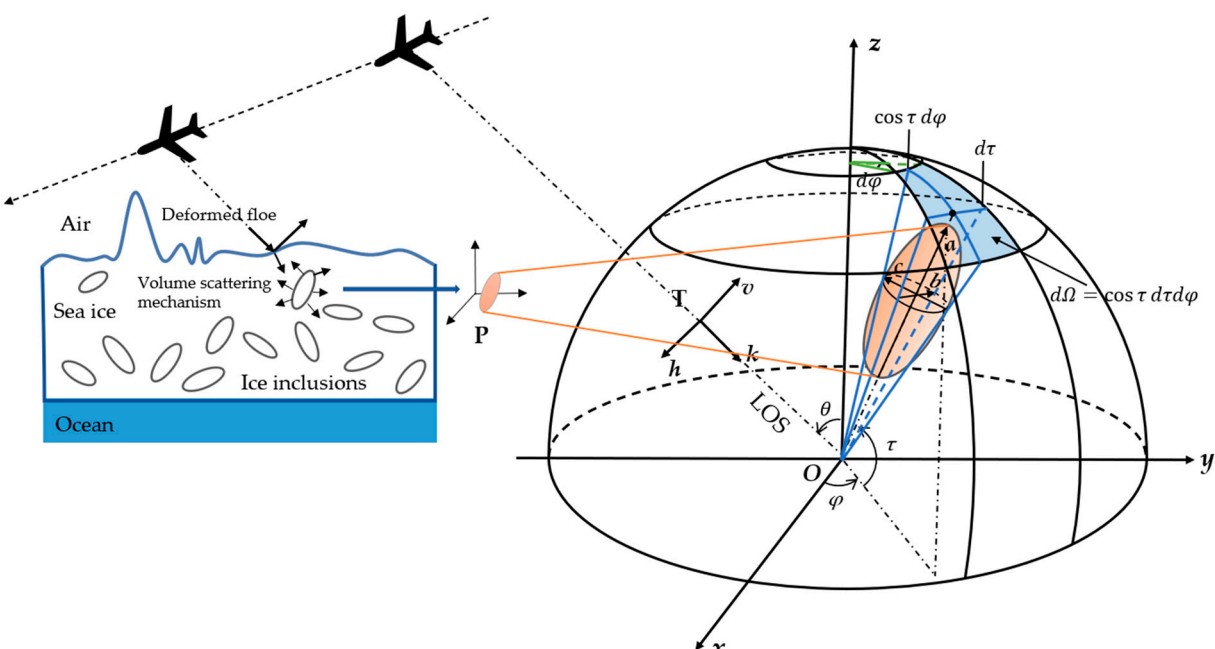

**Figure 1.** Volume scattering mechanism induced by the 3D-oriented brine inclusions within sea ice, and the Cartesian, polarization, ellipsoidal and solid angle geometry in modeling of the polarimetric backscattering and angular distribution of 3D-oriented particles.

### 2.1. Scattering Matrix for a 3D Oriented Ellipsoidal Particle

As illustrated in Figure 1, the volume scattering modeling starts by establishing the polarimetric scattering field of the ellipsoidal ice particle **P** illuminated by the radar transmitted wave **T**. To facilitate the description of the 3D geometry of the particle and radar, three reference coordinate systems are constructed: the Earth-based Cartesian coordinate system $[x, y, z]$ located at **O**, where $z$ is normal to the local surface and $x$ and $y$ are parallel to the azimuth and ground-range directions of radar; the radar polarization coordinate system $[h, k, v]$, where $k$ is the propagation direction of incident wave, i.e., the LOS of radar, with incident angle $\theta \in \left[0, \frac{\pi}{2}\right]$, and $h$ and $v$ are the horizontal and vertical polarization directions of the incident electric field; and the ellipsoid coordinate system $[a, b, c]$, where

*a*, *b* and *c* represent the three axes of the ellipsoidal particle. Furthermore, to simplify the modeling, the LOS *k* of radar is assumed within the plane $(y, z)$, and *h* is aligned with *x*. As a result, $[x, y, z]$ is then a $\left(\frac{\pi}{2} - \theta\right)$-rotation of $[h, k, v]$ around *h*:

$$\begin{bmatrix} x \\ y \\ z \end{bmatrix} = [R_\theta] \begin{bmatrix} h \\ k \\ v \end{bmatrix}, [R_\theta] = \begin{bmatrix} 1 & 0 & 0 \\ 0 & \cos\left(\frac{\pi}{2} - \theta\right) & \sin\left(\frac{\pi}{2} - \theta\right) \\ 0 & -\sin\left(\frac{\pi}{2} - \theta\right) & \cos\left(\frac{\pi}{2} - \theta\right) \end{bmatrix} \tag{1}$$

Instead of the 2D orientation around *k*, the 3D target orientation is determined by the two Euler angles $\varphi$ and $\tau$, where the canting angle $\varphi \in [-\pi, \pi]$ is defined as the rotation about *z*; the tilt angle $\tau \in \left[-\frac{\pi}{2}, \frac{\pi}{2}\right]$ is the subsequent rotation about *y*. Since $\varphi$ and $\tau$ represent two different Euler geometries, without loss of generality, they are assumed to be independent of each other in this article. The relationship between $[x, y, z]$ and $[a, b, c]$ can be expressed in terms of the Euler transform, as follows:

$$\begin{bmatrix} a \\ b \\ c \end{bmatrix} = [R_\tau][R_\varphi] \begin{bmatrix} x \\ y \\ z \end{bmatrix}, [R_\tau] = \begin{bmatrix} \cos\tau & 0 & \sin\tau \\ 0 & 1 & 0 \\ -\sin\tau & 0 & \cos\tau \end{bmatrix}, [R_\varphi] = \begin{bmatrix} \cos\varphi & \sin\varphi & 0 \\ -\sin\varphi & \cos\varphi & 0 \\ 0 & 0 & 1 \end{bmatrix} \tag{2}$$

As a result, the transformation from radar geometry to particle geometry is as follows:

$$\begin{bmatrix} a \\ b \\ c \end{bmatrix} = [R] \begin{bmatrix} h \\ k \\ v \end{bmatrix}, [R] = [R_\tau][R_\varphi][R_\theta] \tag{3}$$

Based on the rotation matrix $[R]$, we can then model the polarizability tensor of the non-chiral ellipsoidal particle in the radar coordinate system as:

$$[\bar{\bar{\alpha}}_r] = [R]^{\mathrm{T}}[\bar{\bar{\alpha}}_p][R], [\bar{\bar{\alpha}}_p] = \begin{bmatrix} \rho_a & & \\ & \rho_b & \\ & & \rho_c \end{bmatrix} \tag{4}$$

where $[\bar{\bar{\alpha}}_r]$ and $[\bar{\bar{\alpha}}_p]$ are the polarizability tensors in radar and particle coordinate systems, respectively. The superscript T denotes the matrix transpose, $\rho_a$, $\rho_b$ and $\rho_c$ are the particle polarizabilities in the ellipsoidal axes *a*, *b* and *c*, respectively, offering a wide variation in particle shapes from plate $(\rho_a = 0)$ to oblate $(\rho_a < \rho_b = \rho_c)$, sphere $(\rho_a = \rho_b = \rho_c)$, prolate $(\rho_a > \rho_b = \rho_c)$ and finally dipole $(\rho_b = \rho_c = 0)$ [46,49]. Based on $[\bar{\bar{\alpha}}_r]$ and the theory of small particle scattering [61], the electric dipole moment induced by the incident field in the particle can be expressed as:

$$\begin{bmatrix} p_h \\ p_k \\ p_v \end{bmatrix} = [\bar{\bar{\alpha}}_r] \begin{bmatrix} E_h^{\mathrm{inc}} \\ 0 \\ E_v^{\mathrm{inc}} \end{bmatrix} \tag{5}$$

where $p_h$, $p_k$ and $p_v$ are the components of the electric moment in the coordinate system $[h, k, v]$; $E_h^{\mathrm{inc}}$ and $E_v^{\mathrm{inc}}$ are the horizontal and vertical entries of the incident electric field. As a result, the backscatter field at the receiver position generated by the electric dipole moment in the convention of BSA is explicitly given as follows [61]:

$$\begin{bmatrix} E_h^{\mathrm{sct}} \\ 0 \\ E_v^{\mathrm{sct}} \end{bmatrix} = \frac{jkZ_0 e^{-jkr}}{4\pi r} \begin{bmatrix} p_h \\ 0 \\ p_v \end{bmatrix} \tag{6}$$

where *k* is the wave number; $Z_0$ is the wave impedance of medium; *r* denotes the distance between particle and receiver; $E_h^{\mathrm{sct}}$ and $E_v^{\mathrm{sct}}$ are the horizontal and vertical components of

the backscatter electric field, which are related to the incident field by the scattering matrix $[S_P]$ of the 3D oriented ellipsoidal particle **P**:

$$
\begin{bmatrix} E_h^{\mathrm{sct}} \\ E_v^{\mathrm{sct}} \end{bmatrix} = \frac{e^{-jkr}}{\sqrt{4\pi r}} [S_P] \begin{bmatrix} E_h^{\mathrm{inc}} \\ E_v^{\mathrm{inc}} \end{bmatrix} \tag{7}
$$

Then, based on Equations (4) to (7), we can easily determine that:

$$
[S_P(\theta, \varphi, \tau)] = \begin{bmatrix} S_{HH} & S_{HV} \\ S_{VH} & S_{VV} \end{bmatrix} = \frac{jkZ_0}{\sqrt{4\pi}} [R_c]^{\mathrm{T}} [\bar{\bar{\alpha}}_p] [R_c], [R_c] = [R] \begin{bmatrix} 1 & 0 \\ 0 & 0 \\ 0 & 1 \end{bmatrix} \tag{8a}
$$

If we neglect the target-independent factor $\frac{jkZ_0}{\sqrt{4\pi}}$, each entry of $[S_P]$ is then directly achieved by combining Equations (1)–(4) into Equation (8a):

$$
S_{HH} = \rho_a \cos^2\tau \cos^2\varphi + \rho_b \sin^2\varphi + \rho_c \sin^2\tau \cos^2\varphi \tag{8b}
$$

$$
\begin{aligned}
S_{VV} &= \rho_a (\cos\tau\sin\varphi\cos\theta + \sin\tau\sin\theta)^2 \\
&\quad + \rho_b \cos^2\varphi\cos^2\theta + \rho_c(\sin\tau\sin\varphi\cos\theta - \cos\tau\sin\theta)^2
\end{aligned} \tag{8c}
$$

$$
\begin{aligned}
S_{HV} = S_{VH} &= \rho_a \cos\tau\cos\varphi(\cos\tau\sin\varphi\cos\theta + \sin\tau\sin\theta) \\
&\quad - \rho_b \cos\varphi\sin\varphi\cos\theta + \rho_c\sin\tau\cos\varphi(\sin\tau\sin\varphi\cos\theta - \cos\tau\sin\theta)
\end{aligned} \tag{8d}
$$

It is noted that, for the 3D geometry shown in Figure 1, Parrella et al. [46,49] have obtained a similar result as that described by Equations (8b)–(8d), among which the expression of $S_{HH}$ and $S_{HV}$ is consistent with Equations (8b)–(8d).However, in contrast with Equation (8c), a typo exists in [46,49], missing the square operation in the first and third terms on the right-hand side of the expression of $S_{VV}$. Equations (8b)–(8d) will become Neumann et al.'s coherent modeling of an oriented spheroid [52] when $\theta = 0$, $\varphi = \psi$, $\tau = \frac{\pi}{2} - \nu$, $\rho_a = \alpha_a$ and $\rho_b = \rho_c = \alpha_b$.

### 2.2. Coherency Matrix for a 3D Oriented Spheroidal Particle

A widely-used approximation in the volume scattering modeling of vegetation cover, as well as soil and ice inclusions, is to consider spheroidal particles with equal minor axes [46,49], i.e., $\rho_b = \rho_c$. Then the Pauli vector $\boldsymbol{k}_P$ of the 3D oriented spheroidal particle is attained by reformulating Equations (8a)–(8d) as

$$
\boldsymbol{k}_P = \frac{1}{\sqrt{2}} \begin{bmatrix} 1 & 0 & 0 \\ 0 & \cos 2\gamma & \sin 2\gamma \\ 0 & -\sin 2\gamma & \cos 2\gamma \end{bmatrix} \begin{bmatrix} \rho_\Delta\left(1 - (\sin\tau\cos\theta - \cos\tau\sin\varphi\sin\theta)^2\right) + 2\rho_b \\ \rho_\Delta\left(1 - (\sin\tau\cos\theta - \cos\tau\sin\varphi\sin\theta)^2\right) \\ 0 \end{bmatrix} \tag{9a}
$$

where:

$$
\gamma = \tan^{-1}\left(\frac{\cos\tau\sin\varphi\cos\theta + \sin\tau\sin\theta}{\cos\tau\cos\varphi}\right), \rho_\Delta = \rho_a - \rho_b \tag{9b}
$$

$\gamma$ denotes the orientation of the particle around the radar LOS, since Equation (9a) is coherent with the scattering vector model $e_T^{SV}$ proposed by Touzi [21]. $\gamma$ will be equal to the canting angle $\varphi$ if the contribution of incidence is neglected (i.e., $\theta = 0$); then the expression of $\boldsymbol{k}_P$ in Equation (9a) will change to that of Equation (3.87) in Cloude [61]. Based on $\boldsymbol{k}_P$, the second-order descriptor, i.e., the coherency matrix $[T_P]$ of the particle is achieved by:

$$
[T_P(\theta, \varphi, \tau)] = \boldsymbol{k}_p \boldsymbol{k}_p^{\mathbf{H}} = \begin{bmatrix} T_{P11} & T_{P12} & T_{P13} \\ T_{P21} & T_{P22} & T_{P23} \\ T_{P31} & T_{P32} & T_{P33} \end{bmatrix} \tag{10a}
$$

where superscript **H** indicates the operator of the conjugate transpose, and $T_{Pmn}$ is the $(m, n)$ element of $[T_P]$ $(m, n = 1, 2, 3)$:

$$T_{P11} = \frac{1}{2}\left(\rho_\Sigma - \rho_\Delta(\sin\tau\cos\theta - \cos\tau\sin\varphi\sin\theta)^2\right)^2, \rho_\Sigma = \rho_a + \rho_b \tag{10b}$$

$$T_{P12} = T_{P21} = \frac{1}{2}\rho_\Delta\left(\rho_\Sigma - \rho_\Delta(\sin\tau\cos\theta - \cos\tau\sin\varphi\sin\theta)^2\right)\left(\cos^2\tau\cos^2\varphi - (\cos\tau\sin\varphi\cos\theta + \sin\tau\sin\theta)^2\right) \tag{10c}$$

$$T_{P13} = T_{P31} = \rho_\Delta\cos\tau\cos\varphi\left(\rho_\Sigma - \rho_\Delta(\sin\tau\cos\theta - \cos\tau\sin\varphi\sin\theta)^2\right)(\cos\tau\sin\varphi\cos\theta + \sin\tau\sin\theta) \tag{10d}$$

$$T_{P22} = \frac{1}{2}\rho_\Delta^2\left(\cos^2\tau\cos^2\varphi - (\cos\tau\sin\varphi\cos\theta + \sin\tau\sin\theta)^2\right)^2 \tag{10e}$$

$$T_{P23} = T_{P32} = \rho_\Delta^2\cos\tau\cos\varphi(\cos\tau\sin\varphi\cos\theta + \sin\tau\sin\theta)\left(\cos^2\tau\cos^2\varphi - (\cos\tau\sin\varphi\cos\theta + \sin\tau\sin\theta)^2\right) \tag{10f}$$

$$T_{P33} = 2\rho_\Delta^2\cos^2\tau\cos^2\varphi(\cos\tau\sin\varphi\cos\theta + \sin\tau\sin\theta)^2 \tag{10g}$$

Like the elements of the scattering matrix $[S_P]$ in Equations (8b)–(8d), all the elements of the coherency matrix $[T_P]$ in Equations (10b)–(10g) are real, because the ellipsoidal particle in the modeling can be generally considered to be made from the non-chiral materials [61]. Although the dimension of $[T_P]$ is higher than that of $[S_P]$, they possess the same five DoF when describing the scattering of a spheroidal particle. Like $[S_P]$, $[T_P]$ in Equations (10b)–(10g) is also determined by the 3D orientation $(\varphi, \tau)$ of the particle. A change in $(\varphi, \tau)$ will lead to a different $[T_P]$, even though the particles present identical shape and composition toward a constant-incidence radar. To model the incoherent scattering behavior of a large volume of such differently oriented particles, a volume coherency matrix $[T_V]$ is often constructed by ensemble averaging the elemental $(\varphi, \tau)$-related matrix $[T_P]$ over the effective ranges of $\varphi$ and $\tau$ [40,46,49], as presented in the following two sections.

### 3. PDFs Modeling for 3D Uniformly Oriented Ellipsoidal Particles

The ensemble averaging relative to variables $\varphi$ and $\tau$ is closely related to the joint PDF $p(\varphi, \tau)$. We have $p(\varphi, \tau) = p(\varphi)p(\tau)$ in view of the independence of $\varphi$ and $\tau$. Theoretically, any choice of $p(\tau)$ and $p(\varphi)$ is feasible. In practice, however, the orientations of ice particles tend to be uniformly distributed in 3D space [40,45,46,49]. i.e., the sea ice particles are uniformly oriented in all directions. To obtain this UOD, Sharma et al. [40] assumed angles $\varphi$ and $\tau$ a uniform distribution over their respective ranges and introduced a factor $\cos\tau$ in the ensemble averaging. All these factors are equivalent to the use of the following PDFs if we combine the factor $\cos\tau$ and uniform $p(\tau)$:

$$\begin{cases} p(\varphi) = \frac{1}{2\Delta\varphi}, \Delta\varphi \in (0, \pi] \\ p(\tau) = \frac{\cos\tau}{2\Delta\tau}, \Delta\tau \in \left(0, \frac{\pi}{2}\right] \end{cases} \tag{11}$$

These PDFs have also been used in the volume modeling of Parrella et al. [46,49]. However, based on the following derivation, one can observe that the PDFs in Equation (11) cannot always ensure the real UOD of particles. UOD is only obtained by distributing the solid angles of particles uniformly in 3D space.

#### 3.1. Solid Angle

As shown in Figure 1, Euler angles $(\varphi, \tau)$ define the 3D orientation of the ellipsoidal particle **P** by determining its major axis $a$. From the viewpoint of **O**, the two minor axes $b$ and $c$ of **P** form two angles $d\varphi$ and $d\tau$ with the common vertex **O**, respectively. The values of $d\varphi$ and $d\tau$, i.e., $d\varphi$ and $d\tau$, are considered to be infinitesimal, since **P** is within the far field of the radar. A minimal solid angle $d\Omega$ is then uniquely constructed by $d\varphi$ and $d\tau$, with **O** as the apex and $a$ as the central axis. Obviously, $d\Omega$ presents the same 3D

orientation as **P**. Therefore, the 3D uniform distribution of orientation can be established directly from the solid angle. It is easy to obtain the value of $d\Omega$ from Figure 1:

$$d\Omega = \cos\tau d\tau d\varphi \tag{12}$$

Then, the total solid angle $\Omega_t$ enclosed by all the 3D orientations of particles is achieved by:

$$\Omega_t = \int_{\mathcal{R}_\Omega} d\Omega = \int_{\mathcal{R}_\varphi}\int_{\mathcal{R}_\tau} \cos\tau d\tau d\varphi \tag{13}$$

where $\mathcal{R}_\Omega \overset{\text{def}}{=} [0, \Omega_t]$ is the effective range of solid angle; $\mathcal{R}_\varphi$ and $\mathcal{R}_\tau$ are the effective ranges of $\varphi$ and $\tau$:

$$\mathcal{R}_\varphi(\varphi_0, \Delta\varphi) \overset{\text{def}}{=} [\varphi_0 - \Delta\varphi, \varphi_0 + \Delta\varphi], \mathcal{R}_\tau(\tau_0, \Delta\tau) \overset{\text{def}}{=} [\tau_0 - \Delta\tau, \tau_0 + \Delta\tau] \tag{14a}$$

$\Delta\varphi$ and $\Delta\tau$ are the half widths of $\mathcal{R}_\varphi$ and $\mathcal{R}_\tau$, respectively:

$$\Delta\varphi \in [0, \pi], \Delta\tau \in \left[0, \frac{\pi}{2}\right] \tag{14b}$$

with $\varphi_0$ and $\tau_0$ being the interval centers:

$$\varphi_0 \in \mathcal{R}_\varphi^m \overset{\text{def}}{=} [-\pi, \pi] = \mathcal{R}_\varphi(0, \pi), \tau_0 \in \mathcal{R}_\tau^m \overset{\text{def}}{=} \left[-\frac{\pi}{2}, \frac{\pi}{2}\right] = \mathcal{R}_\tau\left(0, \frac{\pi}{2}\right) \tag{14c}$$

$\mathcal{R}_\varphi^m$ and $\mathcal{R}_\tau^m$ represent the maximum effective ranges of $\varphi$ and $\tau$, respectively.

The double definite integral for variables $\varphi$ and $\tau$ in Equation (13) is not direct, as it involves the relation between $\mathcal{R}_\varphi$ and $\mathcal{R}_\varphi^m$, as well as that between $\mathcal{R}_\tau$ and $\mathcal{R}_\tau^m$. Here, we first explore the relation between the effective range $\mathcal{R}_\tau$ and the maximum effective range $\mathcal{R}_\tau^m$ of tilt $\tau$. We obtain $\mathcal{R}_\tau \subseteq \mathcal{R}_\tau^m$ when $\Delta\tau \in \left[0, \frac{\pi}{2} - |\tau_0|\right]$. This relation, however, is broken when $\Delta\tau \in \left(\frac{\pi}{2} - |\tau_0|, \frac{\pi}{2}\right]$, because a portion of $\mathcal{R}_\tau$ are out of $\mathcal{R}_\tau^m$ in this case. Let us take $\tau_0 = \frac{\pi}{3}, \Delta\tau = \frac{\pi}{4}$, for instance; here, we obtain $\mathcal{R}_\tau = \left[\frac{\pi}{12}, \frac{7\pi}{12}\right] = \left[\frac{\pi}{12}, \frac{\pi}{2}\right] \cup \left[\frac{\pi}{2}, \frac{7\pi}{12}\right]$, while $\left[\frac{\pi}{2}, \frac{7\pi}{12}\right]$ is out of $\mathcal{R}_\tau^m$. Nevertheless, it can be observed from Figure 1 that all the angles $\tau \pm k\pi$ ($k$ is any integer) correspond to the same tilt geometry; in other words, the period of $\tau$ is $\pi$ in geometry. As a result, $\left[\frac{\pi}{2}, \frac{7\pi}{12}\right]$ is geometrically equivalent to $\left[-\frac{\pi}{2}, -\frac{5\pi}{12}\right] \subset \mathcal{R}_\tau^m$, then $\left[\frac{\pi}{12}, \frac{7\pi}{12}\right] \equiv \left[-\frac{\pi}{2}, -\frac{5\pi}{12}\right] \cup \left[\frac{\pi}{12}, \frac{\pi}{2}\right]$, i.e., $\mathcal{R}_\tau$ is equivalently composed of two disjoint subintervals within $\mathcal{R}_\tau^m$. This also equals subtracting the complementary interval of $\mathcal{R}_\tau$ from $\mathcal{R}_\tau^m$, i.e., $\left[\frac{\pi}{12}, \frac{7\pi}{12}\right] \equiv \left[-\frac{\pi}{2}, \frac{\pi}{2}\right] - \left(-\frac{5\pi}{12}, \frac{\pi}{12}\right)$. We formulate these two cases of $\mathcal{R}_\tau$ as follows:

$$\mathcal{R}_\tau \equiv \begin{cases} \mathcal{R}_\tau^h(\tau_0, \Delta\tau) = [\tau_0 - \Delta\tau, \tau_0 + \Delta\tau] & , \Delta\tau \in \left[0, \frac{\pi}{2} - |\tau_0|\right] \\ \mathcal{R}_\tau^v(\tau_0, \Delta\tau) = \mathcal{R}_\tau^m - \left(\tau_0^c - \Delta\tau^c, \tau_0^c + \Delta\tau^c\right), \Delta\tau \in \left(\frac{\pi}{2} - |\tau_0|, \frac{\pi}{2}\right] \end{cases}, \tau_0 \in \mathcal{R}_\tau^m \tag{15a}$$

The subscripts $h$ and $v$ are introduced to distinguish the two cases of $\mathcal{R}_\tau$ for they roughly correspond to the geometry of horizontal tilt and vertical tilt, where:

$$\tau_0^c = \tau_0 - \text{sgn}(\tau_0)\frac{\pi}{2}, \Delta\tau^c = \frac{\pi}{2} - \Delta\tau \tag{15b}$$

$\text{sgn}(\cdot)$ denotes the sign function. As a result, if we define:

$$S_{2k-1}^\tau = \frac{\int_{\mathcal{R}_\tau} \sin(2k-1)\tau d\tau}{\int_{\mathcal{R}_\tau} \cos\tau d\tau}, C_{2k-1}^\tau = \frac{\int_{\mathcal{R}_\tau} \cos(2k-1)\tau d\tau}{\int_{\mathcal{R}_\tau} \cos\tau d\tau}, k = 1, 2, 3\cdots \tag{16a}$$

from Equation (15) we then have:

$$
\begin{cases}
S_{2k-1}^{\tau h} = \dfrac{\int_{\mathcal{R}_\tau^h} \sin(2k-1)\tau d\tau}{\int_{\mathcal{R}_\tau^h} \cos\tau d\tau} = \dfrac{\sin(2k-1)\tau_0 \mathrm{sinc}(2k-1)\Delta\tau}{\cos\tau_0 \mathrm{sinc}\Delta\tau} \\[4mm]
C_{2k-1}^{\tau h} = \dfrac{\int_{\mathcal{R}_\tau^h} \cos(2k-1)\tau d\tau}{\int_{\mathcal{R}_\tau^h} \cos\tau d\tau} = \dfrac{\cos(2k-1)\tau_0 \mathrm{sinc}(2k-1)\Delta\tau}{\cos\tau_0 \mathrm{sinc}\Delta\tau}
\end{cases}
\tag{16b}
$$

$$
\begin{cases}
S_{2k-1}^{\tau v} = \dfrac{\int_{\mathcal{R}_\tau^v} \sin(2k-1)\tau d\tau}{\int_{\mathcal{R}_\tau^v} \cos\tau d\tau} = \dfrac{\mathrm{sgn}(\tau_0)\cos(2k-1)\tau_0 \cos(2k-1)\Delta\tau}{(2k-1)(1-\sin|\tau_0|\cos\Delta\tau)} \\[4mm]
C_{2k-1}^{\tau v} = \dfrac{\int_{\mathcal{R}_\tau^v} \cos(2k-1)\tau d\tau}{\int_{\mathcal{R}_\tau^v} \cos\tau d\tau} = \dfrac{(-1)^{k-1}-\sin((2k-1)|\tau_0|)\cos(2k-1)\Delta\tau}{(2k-1)(1-\sin|\tau_0|\cos\Delta\tau)}
\end{cases}
\tag{16c}
$$

Likewise, the discontinuity on $\mathcal{R}_\varphi$ is presented as follows:

$$
\mathcal{R}_\varphi \equiv
\begin{cases}
\mathcal{R}_\varphi^s = [\varphi_0 - \Delta\varphi, \varphi_0 + \Delta\varphi] & ,\Delta\varphi \in [0, \pi - |\varphi_0|] \\[2mm]
\mathcal{R}_\varphi^a = \mathcal{R}_\varphi^m - (\varphi_0^c - \Delta\varphi^c, \varphi_0^c + \Delta\varphi^c), \Delta\varphi \in (\pi - |\varphi_0|, \pi]
\end{cases}
, \varphi_0 \in \mathcal{R}_\varphi^m
\tag{17a}
$$

where:

$$
\varphi_0^c = \varphi_0 - \mathrm{sgn}(\varphi_0)\pi, \Delta\varphi^c = \pi - \Delta\varphi
\tag{17b}
$$

Nevertheless, from Figure 1 we observe that the angles $\varphi \pm 2k\pi$ correspond to the same canting geometry, which show the same period as the trigonometric functions. As a result, the discontinuity in Equation (17a) does not affect the results of the following integrals:

$$
\begin{cases}
S_k^\varphi = \dfrac{\int_{\mathcal{R}_\varphi} \sin k\varphi d\varphi}{\int_{\mathcal{R}_\varphi} d\varphi} = \dfrac{\int_{\mathcal{R}_\varphi^s} \sin k\varphi d\varphi}{\int_{\mathcal{R}_\varphi^s} d\varphi} = \dfrac{\int_{\mathcal{R}_\varphi^a} \sin k\varphi d\varphi}{\int_{\mathcal{R}_\varphi^a} d\varphi} = \cos k\varphi_0 \mathrm{sinc} k\Delta\varphi \\[4mm]
C_k^\varphi = \dfrac{\int_{\mathcal{R}_\varphi} \cos k\varphi d\varphi}{\int_{\mathcal{R}_\varphi} d\varphi} = \dfrac{\int_{\mathcal{R}_\varphi^s} \cos k\varphi d\varphi}{\int_{\mathcal{R}_\varphi^s} d\varphi} = \dfrac{\int_{\mathcal{R}_\varphi^a} \cos k\varphi d\varphi}{\int_{\mathcal{R}_\varphi^a} d\varphi} = \sin k\varphi_0 \mathrm{sinc} k\Delta\varphi
\end{cases}
\tag{18}
$$

Combine Equation (16), the integral in Equation (13) is then solved:

$$
\Omega_t =
\begin{cases}
\int_{\mathcal{R}_\varphi} \int_{\mathcal{R}_\tau^h} \cos\tau d\tau d\varphi = 4\Delta\varphi\cos\tau_0\sin\Delta\tau & ,\Delta\tau \in \left[0, \frac{\pi}{2} - |\tau_0|\right] \\[2mm]
\int_{\mathcal{R}_\varphi} \int_{\mathcal{R}_\tau^v} \cos\tau d\tau d\varphi = 4\Delta\varphi(1 - |\sin\tau_0|\cos\Delta\tau), \Delta\tau \in \left(\frac{\pi}{2} - |\tau_0|, \frac{\pi}{2}\right]
\end{cases}
, \Delta\varphi \in [0, \pi]
\tag{19}
$$

This provides a general formula for computing the solid angle over any canting-tilt grid cell. The two branches in Equation (19) obtain the same result $\Omega_t = 4\pi$ when $\Delta\varphi = \pi$ and $\Delta\tau = \frac{\pi}{2}$, which is the solid angle of a sphere.

### 3.2. PDFs

We define $p(\Omega)$ as the PDF of the solid angle within $\mathcal{R}_\Omega$. It is then evident that UOD is achieved if $p(\Omega)$ satisfies:

$$
p(\Omega) = \frac{1}{\Omega_t} =
\begin{cases}
\frac{1}{4\Delta\varphi\cos\tau_0\sin\Delta\tau} & ,\Delta\tau \in \left(0, \frac{\pi}{2} - |\tau_0|\right] \\[2mm]
\frac{1}{4\Delta\varphi(1-|\sin\tau_0|\cos\Delta\tau)}, \Delta\tau \in \left(\frac{\pi}{2} - |\tau_0|, \frac{\pi}{2}\right]
\end{cases}
, \Delta\varphi \in (0, \pi]
\tag{20}
$$

Let $p(\varphi, \tau)$ be the joint PDF of $\varphi$ and $\tau$. According to Equation (12) and the normalization property of PDF we have:

$$
\int_{\mathcal{R}_\Omega} p(\Omega)d\Omega = \int_{\mathcal{R}_\varphi} \int_{\mathcal{R}_\tau} p(\Omega)\cos\tau d\tau d\varphi = \int_{\mathcal{R}_\varphi} \int_{\mathcal{R}_\tau} p(\varphi, \tau)d\tau d\varphi = 1
\tag{21}
$$

An obvious solution to Equation (21) is:

$$
p(\Omega)\cos\tau = p(\varphi, \tau) = p(\varphi)p(\tau)
\tag{22}
$$

Based on Equation (22) and the uniform $p(\Omega)$ in Equation (20), the $p(\varphi)$ and $p(\tau)$ for UOD are finally obtained:

$$\begin{cases} p(\varphi) = \frac{1}{2\Delta\varphi} & , \Delta\varphi \in (0, \pi] \\ p(\tau) = \begin{cases} p_h(\tau) = \frac{\cos\tau}{2\cos\tau_0\sin\Delta\tau} & , \Delta\tau \in \left(0, \frac{\pi}{2} - |\tau_0|\right] \\ p_v(\tau) = \frac{\cos\tau}{2(1-|\sin\tau_0|\cos\Delta\tau)}, \Delta\tau \in \left(\frac{\pi}{2} - |\tau_0|, \frac{\pi}{2}\right] \end{cases} \end{cases} \tag{23a}$$

Equation (23a) formulates the same $p(\varphi)$ as Equation (11), but also the improved $p(\tau)$ for 3D uniform modeling of ice particle orientation. The two branches of $p(\tau)$ in Equation (23a) are complementary, which roughly relates to the tilt geometry of horizontal preference and vertical preference, respectively. They are denoted as $p_h(\tau)$ and $p_v(\tau)$ for convenience. The two branches intersect at $\Delta\tau = \frac{\pi}{2}$, and we obtain $p_h(\tau) = p_v(\tau) = \frac{\cos\tau}{2}$. This shows a randomly 3D uniform distribution without any orientation preference, and we denote it as $p_r(\tau)$. A similar PDF has been used by Nghiem et al. [52] to depict random orientation of scatterers such as ice grains in snow. It was also used by Cloude [60] to characterize the volume scattering and depolarization of particles with random distribution of orientation. The PDF Yamaguchi et al. [23] used to model the orientation $\theta$ of dipole clouds around radar LOS is also $p(\theta) = \frac{\cos\theta}{2}$.

Nevertheless, as an exception, $p(\tau)$ in Equation (23a) is invalid if all the particles bear the single tilt $\tau_0$ with $\Delta\tau = 0$. This also happens to $p(\varphi)$ if particles orient the same canting $\varphi_0$ with $\Delta\varphi = 0$. To fix this, in view of the normalization property of PDF, we define:

$$\begin{cases} p(\varphi) = \delta(\varphi - \varphi_0), \Delta\varphi = 0 \\ p(\tau) = \delta(\tau - \tau_0), \Delta\tau = 0 \end{cases} \tag{23b}$$

where $\delta(\cdot)$ is the Dirac function. In fact, a similar PDF $p(\psi) = \delta(\psi)$ has been proposed by Nghiem et al. [52] to depict brine inclusions oriented in vertical direction. Like $\tau$, here $\psi$ also denotes tilt angle, but $\psi = \frac{\pi}{2} - \tau$. As a result, $p(\psi) = \delta(\psi)$ is equivalent to $p(\tau) = \delta(\tau - \tau_0)$ with $\tau_0 = \frac{\pi}{2}$, which has also been used by Zhang et al. [45] in scattering modeling of the vertically tilted brine inclusions. In their characterization for polarimetric scattering from vegetation canopies, Arii et al. [55] also defined a similar PDF for cases in which all the individual elements have the same orientation. Using Equation (23), a complete expression of the joint PDF $p(\varphi, \tau)$ for UOD with six different cases is finally achieved, as expressed in Equation (24). The six cases are a result of the combination of the three forms of $\Delta\tau$ and the two forms of $\Delta\varphi$. We denote them as Case 1) to Case 6) for convenience. The PDF in Equation (24) will be directly used in the following ensemble modeling of the volume coherency matrix.

$$p(\varphi, \tau) = \begin{cases} \frac{\cos\tau}{4\Delta\varphi\cos\tau_0\sin\Delta\tau} & , \left\{\Delta\tau \in \left(0, \frac{\pi}{2} - |\tau_0|\right], \Delta\varphi \in (0, \pi]\right\} & := \text{Case 1)} \\ \frac{\cos\tau}{4\Delta\varphi(1-|\sin\tau_0|\cos\Delta\tau)} & , \left\{\Delta\tau \in \left(\frac{\pi}{2} - |\tau_0|, \frac{\pi}{2}\right], \Delta\varphi \in (0, \pi]\right\} & := \text{Case 2)} \\ \frac{\delta(\varphi-\varphi_0)\cos\tau}{2\cos\tau_0\sin\Delta\tau} & , \left\{\Delta\tau \in \left(0, \frac{\pi}{2} - |\tau_0|\right], \Delta\varphi = 0\right\} & := \text{Case 3)} \\ \frac{\delta(\varphi-\varphi_0)\cos\tau}{2(1-|\sin\tau_0|\cos\Delta\tau)} & , \left\{\Delta\tau \in \left(\frac{\pi}{2} - |\tau_0|, \frac{\pi}{2}\right], \Delta\varphi = 0\right\} & := \text{Case 4)} \\ \frac{\delta(\tau-\tau_0)}{2\Delta\varphi} & , \left\{\Delta\tau = 0, \Delta\varphi \in (0, \pi]\right\} & := \text{Case 5)} \\ \delta(\varphi - \varphi_0)\delta(\tau - \tau_0) & , \left\{\Delta\tau = 0, \Delta\varphi = 0\right\} & := \text{Case 6)} \end{cases} \tag{24}$$

## 4. Volume Scattering Modeling for 3D Uniformly Distributed Spheroidal Particles

### 4.1. Multiplicative Volume Scattering Model

After establishing the polarimetric coherent scattering and angular distribution of 3D oriented spheroidal particles, we can then construct the volume coherency matrix $[T_V]$ by

ensemble averaging the elemental $(\varphi, \tau)$-dependent coherency matrix $[T_P]$ over the ranges $\mathcal{R}_\varphi$ and $\mathcal{R}_\tau$ to describe the incoherent scattering behavior of a large volume of differently orientated spheroidal particles:

$$[T_V] = \langle [T_P] \rangle = \begin{bmatrix} T_{V11} & T_{V12} & T_{V13} \\ T_{V21} & T_{V22} & T_{V23} \\ T_{V31} & T_{V32} & T_{V33} \end{bmatrix} \tag{25}$$

where $\langle \cdot \rangle$ denotes the operator of ensemble average. Like $[T_P]$, $[T_V]$ is also a symmetric matrix if the particles are made from non-chiral material. $T_{Vmn}$ is the $(m, n)$ element of $[T_V]$:

$$T_{Vmn} = T_{Vnm} = \langle T_{Pmn} \rangle = \int_{\mathcal{R}_\varphi} \int_{\mathcal{R}_\tau} T_{Pmn}(\varphi, \tau) p(\varphi, \tau) d\tau d\varphi, \, m, n = 1, 2, 3 \tag{26a}$$

Bringing Equation (24) into Equation (26a), we then obtain the calculation of $T_{Vmn}$ with six different cases:

$$T_{Vmn} = \begin{cases} \frac{1}{4\Delta\varphi \cos \tau_0 \sin \Delta\tau} \int_{\mathcal{R}_\varphi} \int_{\mathcal{R}_\tau^h} T_{Pmn}(\varphi, \tau) \cos \tau d\tau d\varphi & \text{, Case 1)} \\[2mm] \frac{1}{4\Delta\varphi(1 - |\sin \tau_0| \cos \Delta\tau)} \int_{\mathcal{R}_\varphi} \int_{\mathcal{R}_\tau^v} T_{Pmn}(\varphi, \tau) \cos \tau d\tau d\varphi & \text{, Case 2)} \\[2mm] \frac{1}{2\cos \tau_0 \sin \Delta\tau} \int_{\mathcal{R}_\tau^h} T_{Pmn}(\varphi_0, \tau) \cos \tau d\tau & \text{, Case 3)} \\[2mm] \frac{1}{2(1 - |\sin \tau_0| \cos \Delta\tau)} \int_{\mathcal{R}_\tau^v} T_{Pmn}(\varphi_0, \tau) \cos \tau d\tau & \text{, Case 4)} \\[2mm] \frac{1}{2\Delta\varphi} \int_{\mathcal{R}_\varphi} T_{Pmn}(\varphi, \tau_0) d\varphi & \text{, Case 5)} \\[2mm] T_{Pmn}(\varphi_0, \tau_0) & \text{, Case 6)} \end{cases} \tag{26b}$$

Nevertheless, the six expressions of $T_{Vmn}$ can be perfectly unified into the following form

$$T_{Vmn} = \boldsymbol{\tau}^{\mathrm{T}} [\omega]_{mn} \boldsymbol{\varphi} \tag{27}$$

This enables a multiplicative modeling of the volume scattering of 3D uniformly oriented spheroidal particles. The contributions of radar observation geometry and particle shape are coupled into the matrix $[\omega]_{mn}$, which is a $6 \times 5$ or $6 \times 4$ matrix determined by the radar incidence $\theta$ and particle shape parameters $\rho_\Sigma$ and $\rho_\Delta$:

$$[\omega]_{mn} = [A]_{mn} \bigoplus [B]_{mn} = \begin{bmatrix} [A]_{mn} & \mathbf{0} \\ \mathbf{0} & [B]_{mn} \end{bmatrix}, \, m, n = 1, 2, 3 \tag{28}$$

where $\bigoplus$ indicates the operator of direct sum; $[A]_{mn}$ is a $3 \times 2$ matrix, and $[B]_{mn}$ is a $3 \times 3$ or $3 \times 2$ matrix. The symmetry of $[T_V]$ means that:

$$[\omega]_{mn} = [\omega]_{nm} \leftrightarrow \begin{cases} [A]_{mn} = [A]_{nm} \\ [B]_{mn} = [B]_{nm} \end{cases}, \, m, n = 1, 2, 3 \tag{29}$$

Matrices $[A]_{mn}$ and $[B]_{mn}$ are specific to each upper/lower triangular entry of the volume coherency matrix $[T_V]$, as formulated in Appendix A. In addition to $[\omega]_{mn}$, the 3D orientation of particles contributes to the modeling of volume scattering through the vectors $\boldsymbol{\tau}$ and $\boldsymbol{\varphi}$, as shown in Equation (27), where $\boldsymbol{\tau}$ is a tilt-related vector:

$$\boldsymbol{\tau} = \begin{cases} \boldsymbol{\tau_h} \stackrel{\text{def}}{=} \begin{bmatrix} S_5^{\tau h} & S_3^{\tau h} & S_1^{\tau h} & C_5^{\tau h} & C_3^{\tau h} & 1 \end{bmatrix}^{\mathrm{T}}, \Delta\tau \in \left(0, \frac{\pi}{2} - |\tau_0|\right] \\[2mm] \boldsymbol{\tau_v} \stackrel{\text{def}}{=} \begin{bmatrix} S_5^{\tau v} & S_3^{\tau v} & S_1^{\tau v} & C_5^{\tau v} & C_3^{\tau v} & 1 \end{bmatrix}^{\mathrm{T}}, \Delta\tau \in \left(\frac{\pi}{2} - |\tau_0|, \frac{\pi}{2}\right] \end{cases} \tag{30}$$

and $\boldsymbol{\varphi}$ is a canting-related vector:

$$\boldsymbol{\varphi} = \begin{cases} \boldsymbol{\varphi}_s \stackrel{\text{def}}{=} \begin{bmatrix} S_3^{\varphi} & S_1^{\varphi} & C_4^{\varphi} & C_2^{\varphi} & 1 \end{bmatrix}^{\text{T}} & , m, n = 1, 2 \text{ or } m = n = 3 \\ \boldsymbol{\varphi}_a \stackrel{\text{def}}{=} \begin{bmatrix} C_3^{\varphi} & C_1^{\varphi} & S_4^{\varphi} & S_2^{\varphi} \end{bmatrix}^{\text{T}} & , \text{else} \end{cases} \tag{31}$$

As a result, the volume coherency matrix of particles can be generally expressed as:

$$[T_V] = \begin{bmatrix} \boldsymbol{\tau}^{\text{T}}[\omega]_{11}\boldsymbol{\varphi}_s & \boldsymbol{\tau}^{\text{T}}[\omega]_{12}\boldsymbol{\varphi}_s & \boldsymbol{\tau}^{\text{T}}[\omega]_{13}\boldsymbol{\varphi}_a \\ \boldsymbol{\tau}^{\text{T}}[\omega]_{21}\boldsymbol{\varphi}_s & \boldsymbol{\tau}^{\text{T}}[\omega]_{22}\boldsymbol{\varphi}_s & \boldsymbol{\tau}^{\text{T}}[\omega]_{13}\boldsymbol{\varphi}_a \\ \boldsymbol{\tau}^{\text{T}}[\omega]_{31}\boldsymbol{\varphi}_r & \boldsymbol{\tau}^{\text{T}}[\omega]_{32}\boldsymbol{\varphi}_r & \boldsymbol{\tau}^{\text{T}}[\omega]_{33}\boldsymbol{\varphi}_s \end{bmatrix} = [\Gamma]^{\text{T}}[\Omega][\Phi] \tag{32}$$

where:

$$[\Gamma] = \begin{cases} [\Gamma]_h \stackrel{\text{def}}{=} [I_3] \otimes \boldsymbol{\tau}_h, \Delta\tau \in \left(0, \frac{\pi}{2} - |\tau_0|\right] \\ [\Gamma]_v \stackrel{\text{def}}{=} [I_3] \otimes \boldsymbol{\tau}_v, \Delta\tau \in \left(\frac{\pi}{2} - |\tau_0|, \frac{\pi}{2}\right] \end{cases}, [\Phi] = \begin{bmatrix} [\Phi]_s \stackrel{\text{def}}{=} [I_3] \otimes \boldsymbol{\varphi}_s \\ [\Phi]_a \stackrel{\text{def}}{=} [I_3] \otimes \boldsymbol{\varphi}_a \end{bmatrix} \tag{33}$$

$$[\Omega] = \begin{bmatrix} [\Omega]_s & [\Omega]_r \end{bmatrix}, [\Omega]_s = \begin{bmatrix} [\omega]_{11} & [\omega]_{12} & \mathbf{0} \\ [\omega]_{21} & [\omega]_{22} & \mathbf{0} \\ \mathbf{0} & \mathbf{0} & [\omega]_{33} \end{bmatrix}, [\Omega]_a = \begin{bmatrix} \mathbf{0} & \mathbf{0} & [\omega]_{13} \\ \mathbf{0} & \mathbf{0} & [\omega]_{23} \\ [\omega]_{31} & [\omega]_{32} & \mathbf{0} \end{bmatrix} \tag{34}$$

$[I_3]$ is the $3 \times 3$ identity matrix and $\otimes$ denotes the operator of the Kronecker product. We call $[\Omega]$ the essential matrix, as it is only determined by the particle shape and the observation geometry of the radar. The six volume scattering scenarios in Equation (26b) differ from one another due to the 3D orientations of particles, which is accounted for by the tilt matrix $[\Gamma]$ and canting matrix $[\Phi]$. To investigate these, we first focus on the limit of $[\Gamma]_h$, $[\Gamma]_v$ and $[\Phi]$, and that of $\boldsymbol{\tau}_h$, $\boldsymbol{\tau}_v$ and $\boldsymbol{\varphi}$ when $\Delta\tau$ and $\Delta\varphi$ approach zero. Based on Equations (16b), (16c) and (18) we have:

$$\begin{cases} \lim_{\Delta\tau \to 0} S_{2k-1}^{\tau h} = \frac{\sin{(2k-1)\tau_0}}{\cos\tau_0} \\ \lim_{\Delta\tau \to 0} C_{2k-1}^{\tau h} = \frac{\cos{(2k-1)\tau_0}}{\cos\tau_0} \end{cases}, \begin{cases} \lim_{\Delta\varphi \to 0}\left(\lim_{|\tau_0| \to \frac{\pi}{2}} S_{2k-1}^{\tau v}\right) = 0 \\ \lim_{\Delta\varphi \to 0}\left(\lim_{|\tau_0| \to \frac{\pi}{2}} C_{2k-1}^{\tau v}\right) = (-1)^{k-1}(2k-1) \end{cases}, \begin{cases} \lim_{\Delta\varphi \to 0} S_k^{\varphi} = \cos k\varphi_0 \\ \lim_{\Delta\varphi \to 0} C_k^{\varphi} = \sin k\varphi_0 \end{cases} \tag{35}$$

The limits of tilt matrices $[\Gamma]_h$ and $[\Gamma]_v$ are then obtained as follows:

$$\begin{cases} [\Gamma]_{h0} \stackrel{\text{def}}{=} \lim_{\Delta\tau \to 0}[\Gamma]_h & = [I_3] \otimes \boldsymbol{\tau}_{h0} \\ [\Gamma]_{v0} \stackrel{\text{def}}{=} \lim_{\Delta\tau \to 0}\left(\lim_{|\tau_0| \to \frac{\pi}{2}}[\Gamma]_v\right) & = [I_3] \otimes \boldsymbol{\tau}_{v0} \end{cases} \tag{36a}$$

where:

$$\begin{cases} \boldsymbol{\tau}_{h0} \stackrel{\text{def}}{=} \lim_{\Delta\tau \to 0} \boldsymbol{\tau}_h & = \begin{bmatrix} \frac{\sin 5\tau_0}{\cos\tau_0} & \frac{\sin 3\tau_0}{\cos\tau_0} & \frac{\sin\tau_0}{\cos\tau_0} & \frac{\cos 5\tau_0}{\cos\tau_0} & \frac{\cos 3\tau_0}{\cos\tau_0} & 1 \end{bmatrix}^{\text{T}} \\ \boldsymbol{\tau}_{v0} \stackrel{\text{def}}{=} \lim_{\Delta\tau \to 0}\left(\lim_{|\tau_0| \to \frac{\pi}{2}} \boldsymbol{\tau}_v\right) & = \begin{bmatrix} 0 & 0 & 0 & 5 & -3 & 1 \end{bmatrix}^{\text{T}} \end{cases} \tag{36b}$$

Likewise, the limit of tilt matrix $[\Phi]$ is obtained as follows:

$$[\Phi]_0 = \lim_{\Delta\varphi \to 0}[\Phi] = \begin{bmatrix} [\Phi]_{s0} \stackrel{\text{def}}{=} \lim_{\Delta\varphi \to 0}[\Phi]_s = [I_3] \otimes \boldsymbol{\varphi}_{s0} \\ [\Phi]_{a0} \stackrel{\text{def}}{=} \lim_{\Delta\varphi \to 0}[\Phi]_a = [I_3] \otimes \boldsymbol{\varphi}_{a0} \end{bmatrix} \tag{37a}$$

where:

$$\boldsymbol{\varphi_0} \overset{\text{def}}{=} \lim_{\Delta\varphi\to0} \boldsymbol{\varphi} = \begin{cases} \boldsymbol{\varphi_{s0}} \overset{\text{def}}{=} \lim_{\Delta\varphi\to0} \boldsymbol{\varphi_s} = \begin{bmatrix} \cos 3\varphi_0 & \cos\varphi_0 & \sin 4\varphi_0 & \sin 2\varphi_0 & 1 \end{bmatrix}^{\text{T}} & , m,n = 1,2 \text{ or } m = n = 3 \\ \boldsymbol{\varphi_{a0}} \overset{\text{def}}{=} \lim_{\Delta\varphi\to0} \boldsymbol{\varphi_a} = \begin{bmatrix} \sin 3\varphi_0 & \sin\varphi_0 & \cos 4\varphi_0 & \cos 2\varphi_0 \end{bmatrix}^{\text{T}} & , \text{else} \end{cases} \tag{37b}$$

Based on all these equations, the volume coherency element $T_{Vmn}$ and matrix $[T_V]$ of the six cases can be directly formulated as:

$$T_{Vmn} = \begin{cases} \boldsymbol{\tau}_h^{\text{T}}[\omega]_{mn}\boldsymbol{\varphi} & \leftrightarrow & \text{Case 1)} & \leftrightarrow & [\Gamma]_h^{\text{T}}[\Omega][\Phi] \\ \boldsymbol{\tau}_v^{\text{T}}[\omega]_{mn}\boldsymbol{\varphi} & \leftrightarrow & \text{Case 2)} & \leftrightarrow & [\Gamma]_v^{\text{T}}[\Omega][\Phi] \\ \boldsymbol{\tau}_h^{\text{T}}[\omega]_{mn}\boldsymbol{\varphi_0} & \leftrightarrow & \text{Case 3)} & \leftrightarrow & [\Gamma]_h^{\text{T}}[\Omega][\Phi]_0 \\ \boldsymbol{\tau}_v^{\text{T}}[\omega]_{mn}\boldsymbol{\varphi_0} & \leftrightarrow & \text{Case 4)} & \leftrightarrow & [\Gamma]_v^{\text{T}}[\Omega][\Phi]_0 \\ \boldsymbol{\tau}_{h0}^{\text{T}}[\omega]_{mn}\boldsymbol{\varphi} & \leftrightarrow & \text{Case 5a)} & \leftrightarrow & [\Gamma]_{h0}^{\text{T}}[\Omega][\Phi] \\ \boldsymbol{\tau}_{v0}^{\text{T}}[\omega]_{mn}\boldsymbol{\varphi} & \leftrightarrow & \text{Case 5b)} & \leftrightarrow & [\Gamma]_{v0}^{\text{T}}[\Omega][\Phi] \\ \boldsymbol{\tau}_{h0}^{\text{T}}[\omega]_{mn}\boldsymbol{\varphi_0} & \leftrightarrow & \text{Case 6a)} & \leftrightarrow & [\Gamma]_{h0}^{\text{T}}[\Omega][\Phi]_0 \\ \boldsymbol{\tau}_{v0}^{\text{T}}[\omega]_{mn}\boldsymbol{\varphi_0} & \leftrightarrow & \text{Case 6b)} & \leftrightarrow & [\Gamma]_{v0}^{\text{T}}[\Omega][\Phi]_0 \end{cases} = [T_V] \tag{38}$$

where a dichotomy is carried out for Case 5) and Case 6) according to whether $|\tau_0|$ is $\frac{\pi}{2}$:

$$\begin{cases} \text{Case 5)} := \begin{cases} \text{Case 5a)}, |\tau_0| \neq \frac{\pi}{2} \\ \text{Case 5b)}, |\tau_0| = \frac{\pi}{2} \end{cases}, \Delta\tau = 0, \Delta\varphi \in (0,\pi] \\ \text{Case 6)} := \begin{cases} \text{Case 6a)}, |\tau_0| \neq \frac{\pi}{2} \\ \text{Case 6b)}, |\tau_0| = \frac{\pi}{2} \end{cases}, \Delta\tau = 0, \Delta\varphi = 0 \end{cases} \tag{39}$$

*4.2. Special Scenerios*

Among the six cases listed in Equation (38), Case 6) represents an extreme situation in which all particles are 3D aligned with the same Euler angles $(\varphi_0, \tau_0)$. The scatterings of the particles are completely coherent; the volume coherency matrix $[T_V]$ stays the same as the rank-1 single coherency matrix $[T_P]$:

$$\begin{cases} \Delta\tau = 0 \\ \Delta\varphi = 0 \end{cases} \to [T_V] = [T_P] \leftrightarrow \begin{cases} [\Gamma]_{h0}^{\text{T}}[\Omega][\Phi]_0 = [T_P(\theta, \varphi, \tau = \tau_0 \neq \pm\frac{\pi}{2})] \\ [\Gamma]_{v0}^{\text{T}}[\Omega][\Phi]_0 = [T_P(\theta, \varphi, \tau = \tau_0 = \pm\frac{\pi}{2})] \end{cases} \tag{40}$$

In other cases, the diverse 3D orientation will depolarize the scatterings of particles, which raises the entropy of the volume coherence matrix $[T_V]$ and makes it no longer consistent with $[T_P]$. Nevertheless, the degree of such geometric depolarization is tuned by the shape of the particle and fails when the particles are spheres, i.e., $\rho_\Delta = 0$. Additionally, the observation geometry of radar also loses its influence on $[T_V]$ of spherical particles because $\rho_\Delta = 0$ makes the sub-essential matrices $[\omega]_{mn}$ independent of the radar incidence $\theta$:

$$[\omega]_{mn} = \begin{cases} \frac{\rho_\Sigma^2}{2} \begin{bmatrix} \mathbf{0_{5\times4}} & \mathbf{0_{5\times1}} \\ \mathbf{0_{1\times4}} & 1 \end{bmatrix} & , m = n = 1 \\ \mathbf{0} & , \text{else} \end{cases} \tag{41}$$

As a result, from Equation (38) we can easily determine that:

$$[T_V] = [T_P] = \frac{\rho_\Sigma^2}{2} \begin{bmatrix} 1 & 0 & 0 \\ 0 & 0 & 0 \\ 0 & 0 & 0 \end{bmatrix} \tag{42}$$

This is the canonical model of surface scattering.

From the perspective of tilt geometry, the six cases of $[T_V]$ in Equation (38) correspond to the horizontal tilt and vertical tilt, respectively. For convenience of analysis, we define:

$$\begin{cases} \left[T_V^h\right] = [\Gamma]_h^{\mathrm{T}}[\Omega][\Phi], \tau \in \mathcal{R}_\tau^h(\tau_{h0}, \Delta\tau_h), \tau_{h0} \in \mathcal{R}_\tau^m, \Delta\tau_h \in \left(0, \frac{\pi}{2} - |\tau_{h0}|\right] \\ \left[T_V^v\right] = [\Gamma]_v^{\mathrm{T}}[\Omega][\Phi], \tau \in \mathcal{R}_\tau^v(\tau_{v0}, \Delta\tau_v), \tau_{v0} \in \mathcal{R}_\tau^m, \Delta\tau_v \in \left(\frac{\pi}{2} - |\tau_{v0}|, \frac{\pi}{2}\right] \end{cases} \tag{43}$$

The two tilt situations will intersect when $\Delta\tau_h$ and $\Delta\tau_v$ take the maximum $\frac{\pi}{2}$, because $\Delta\tau = \frac{\pi}{2}$ will make the tilt parameters in Equation (16) independent of $\tau_0$:

$$\Delta\tau = \frac{\pi}{2} \rightarrow \begin{cases} S_{2k-1}^{\tau h} = S_{2k-1}^{\tau v} = S_{2k-1}^{\tau r} = 0 \\ C_{2k-1}^{\tau h} = C_{2k-1}^{\tau v} = C_{2k-1}^{\tau r} = \frac{(-1)^{k-1}}{(2k-1)} \end{cases} \tag{44}$$

As a result, we will have:

$$\boldsymbol{\tau_h} = \boldsymbol{\tau_v} = \boldsymbol{\tau_r} = \begin{bmatrix} 0 & 0 & 0 & \frac{1}{5} & -\frac{1}{3} & 1 \end{bmatrix}^{\mathrm{T}} \tag{45}$$

thus:

$$\left[T_V^h\right] = [T_V^v] = [T_V^r] = \left([I_3] \bigotimes \boldsymbol{\tau_r}\right)^{\mathrm{T}}[\Omega][\Phi] \tag{46}$$

Obviously, $[T_V^r]$ corresponds to a tilt geometry that prefers neither horizontal nor vertical tilt; this is referred to as random tilt. In addition to this equality relationship, we are also interested in the complement tilt (CT) relationship between the two tilt geometries. As illustrated in Figure 2, ranges $\mathcal{R}_\tau^h$ and $\mathcal{R}_\tau^v$ will complement each other when $\tau_{h0}$, $\tau_{v0}$, $\Delta\tau_h$ and $\Delta\tau_v$ satisfy the following condition.

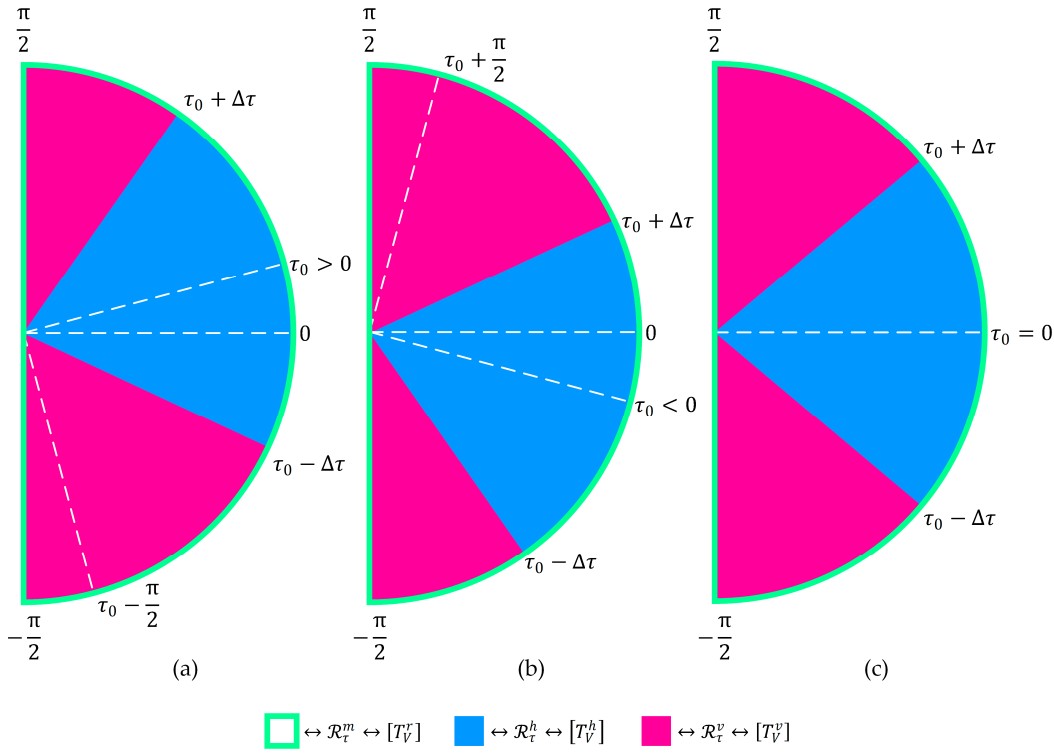

**Figure 2.** The complementary relationship among the geometries of horizontal tilt, vertical tilt and random tilt when (**a**) $\tau_0 > 0$, (**b**) $\tau_0 < 0$ and (**c**) $\tau_0 = 0$.

$$\text{CT} \leftrightarrow \mathcal{R}_\tau^h \bigoplus \mathcal{R}_\tau^v = \mathcal{R}_\tau^m \leftrightarrow \begin{cases} |\tau_{h0} - \tau_{v0}| = \frac{\pi}{2} \\ \Delta\tau_h + \Delta\tau_v = \frac{\pi}{2} \end{cases} \tag{47}$$

For convenience, let $\tau_0 = \tau_{h0}$, $\Delta\tau = \Delta\tau_h$, then the complementary version of tilt parameters in Equation (16) can be expressed as:

$$\text{CT} \rightarrow \begin{cases} S_{2k-1}^{\tau h} = \frac{\sin(2k-1)\tau_0 \sin(2k-1)\Delta\tau}{(2k-1)\cos\tau_0 \sin\Delta\tau} \\ C_{2k-1}^{\tau h} = \frac{\cos(2k-1)\tau_0 \sin(2k-1)\Delta\tau}{(2k-1)\cos\tau_0 \sin\Delta\tau} \end{cases}, \begin{cases} S_{2k-1}^{\tau v} = \frac{-\sin(2k-1)\tau_0 \sin(2k-1)\Delta\tau}{(2k-1)(1-\cos\tau_0 \sin\Delta\tau)} \\ C_{2k-1}^{\tau v} = \frac{(-1)^{k-1} - \cos(2k-1)\tau_0 \sin(2k-1)\Delta\tau}{(2k-1)(1-\cos\tau_0 \sin\Delta\tau)} \end{cases} \tag{48}$$

The following CT relationship on tilt parameters can thus be easily obtained:

$$\text{CT} \rightarrow \begin{cases} \cos\tau_0 \sin\Delta\tau S_{2k-1}^{\tau h} + (1 - \cos\tau_0 \sin\Delta\tau) S_{2k-1}^{\tau v} = S_{2k-1}^{\tau r} \\ \cos\tau_0 \sin\Delta\tau C_{2k-1}^{\tau h} + (1 - \cos\tau_0 \sin\Delta\tau) C_{2k-1}^{\tau v} = C_{2k-1}^{\tau r} \end{cases} \tag{49}$$

From the viewpoint of tilt vectors defined in Equation (30), Equation (49) indicates that:

$$\text{CT} \rightarrow \cos\tau_0 \sin\Delta\tau \boldsymbol{\tau}_h + (1 - \cos\tau_0 \sin\Delta\tau)\boldsymbol{\tau}_v = \boldsymbol{\tau}_r \tag{50}$$

As a result, the similar CT relationship on coherency matrices is finally achieved:

$$\text{CT} \rightarrow \cos\tau_0 \sin\Delta\tau \left[T_V^h\right] + (1 - \cos\tau_0 \sin\Delta\tau)[T_V^v] = [T_V^r] \tag{51}$$

where $\left[T_V^h\right]$, $[T_V^v]$ and $[T_V^r]$ correspond to the tilt ranges of $\mathcal{R}_\tau^h$, $\mathcal{R}_\tau^v$ and $\mathcal{R}_\tau^m$, respectively, which achieves a complete coverage of the tilt geometry from the horizontal, vertical, and random aspects.

Equations (32)–(34) further imply the following dichotomy of matrix $[T_V]$:

$$[T_V] = [T_V^s] + [T_V^a], \begin{cases} [T_V^s] = [\Gamma]^{\text{T}}[\Omega]_s[\Phi]_s = \begin{bmatrix} \times & \times & 0 \\ \times & \times & 0 \\ 0 & 0 & \times \end{bmatrix} \\ [T_V^a] = [\Gamma]^{\text{T}}[\Omega]_a[\Phi]_a = \begin{bmatrix} 0 & 0 & \times \\ 0 & 0 & \times \\ \times & \times & 0 \end{bmatrix} \end{cases} \tag{52}$$

where "$\times$" denotes a non-zero element; $[T_V^s]$ corresponds to a component of reflection symmetry (RS), because it is in the same form as that of a reflection-symmetric media [62]; $[T_V^a]$, on the other hand, shows a residual asymmetric component which disappears when $\Delta\varphi$ takes its maximum $\pi$:

$$\text{RS} \leftrightarrow \Delta\varphi = \pi \rightarrow \begin{cases} S_k^\varphi = 0 \\ C_k^\varphi = 0 \end{cases} \rightarrow \begin{cases} \boldsymbol{\varphi}_s = \begin{bmatrix} \mathbf{0}_{1\times4} & 1 \end{bmatrix}^{\text{T}} \\ \boldsymbol{\varphi}_a = \mathbf{0}_{4\times1} \rightarrow [T_V^a] = [\mathbf{0}_{3\times3}] \end{cases} \rightarrow [T_V] = [T_V^s] \tag{53}$$

$[T_V]$ in this case is fully reflection-symmetric and independent of $\varphi_0$. Hence, if RS is further considered, $\left[T_V^h\right]$ and $[T_V^v]$ will present a similar matrix pattern to $[T_V^s]$ in Equation (52), while $[T_V^r]$ will become the following diagonal matrix of azimuth symmetry (AS) which is independent of $\tau_0$, $\varphi_0$ and $\theta$:

$$\text{AS} \leftrightarrow \begin{cases} \Delta\varphi = \pi \\ \Delta\tau = \frac{\pi}{2} \end{cases} \rightarrow [T_V]_r = \begin{bmatrix} \frac{15\rho_\Sigma^2 - 10\rho_\Delta\rho_\Sigma + 3\rho_\Delta^2}{30} & 0 & 0 \\ 0 & \frac{2\rho_\Delta^2}{15} & 0 \\ 0 & 0 & \frac{2\rho_\Delta^2}{15} \end{bmatrix} \tag{54}$$

Then the CT relationship in Equation (51) immediately shows that:

$$\cos \tau_0 \sin \Delta\tau T_{V12}^h + (1 - \cos \tau_0 \sin \Delta\tau) T_{V12}^v = 0 \rightarrow \text{sgn}\left(T_{V12}^h\right) = -\text{sgn}(T_{V12}^v) \qquad (55)$$

For $[T_V]$ of a volume of differently orientated spheroidal particles constructed in Equation (25), we determine that, in all cases:

$$T_{V12} = \text{Re}(T_{V12}) = \frac{1}{2}\left(\left\langle|S_{HH}|^2\right\rangle - \left\langle|S_{VV}|^2\right\rangle\right) \qquad (56)$$

Therefore, $\text{sgn}(T_{V12})$ is a nice feature to distinguish among $\left[T_V^h\right]$, $\left[T_V^v\right]$ and $\left[T_V^r\right]$.

*4.3. Simulation*

To investigate the performance of the models in characterization of volume behaviors of particles with different shapes under different observation and orientation geometries, following Cloude et al. [60] and Parrella et al. [46], simulations are conducted on a volume of oblate ($\rho_\Sigma = 11, \rho_\Delta = -9$, i.e., $\rho_a = 1, \rho_b = 10$) and prolate ($\rho_\Sigma = 11, \rho_\Delta = 9$, i.e., $\rho_a = 10, \rho_b = 1$) particles with the differential reflectivity $Z_{DR}$, polarimetric entropy $H$ and mean scattering alpha angle $\alpha$ as descriptors. $Z_{DR}$ is the logarithm ratio of the horizontally polarized reflectivity to the vertically polarized reflectivity [63]:

$$Z_{DR} = 10\log_{10}\left(\frac{\left\langle|S_{HH}|^2\right\rangle}{\left\langle|S_{VV}|^2\right\rangle}\right) \qquad (57a)$$

It is a crucial weather radar measurement that helps to identify hail shafts, detect updrafts, determine rain drop size and identify aggregation of dry snow [64]. Additionally, an equivalent definition of $Z_{DR}$ based on coherence matrix $[T]$ is often used in branch condition to select the volume model for four-component model-based compositions [23,39]:

$$Z_{DR}([T]) = 10\log_{10}\left(\frac{T_{11} + T_{22} + 2\text{Re}(T_{12})}{T_{11} + T_{22} - 2\text{Re}(T_{12})}\right) \qquad (57b)$$

Polarization entropy $H$ and mean scattering $\alpha$ angle are two important parameters of the well-known $H/\alpha$ classifier [60]. Their definition on the coherence matrix $[T]$ is as follows:

$$H = -\sum_{i=1}^3 P_i\log_3(P_i), P_i = \frac{\lambda_i}{\sum_{k=1}^3 \lambda_k} \qquad (58)$$

$$\alpha = \sum_{i=1}^3 P_i\cos^{-1}|u_{i1}| \qquad (59)$$

where $\lambda_i$ is the eigenvalue of $[T]$, $u_{i1}$ denotes the first entry of the eigenvector $\boldsymbol{u_i}$, $i = 1, 2, 3$. $H$ depicts the degree of statistical disorder of particle ensemble; $\alpha$ measures the dominant scattering mechanism [62].

Since the brine inclusions within sea ice are usually canted with no preference [45,52], we set $\Delta\varphi = \pi$ in the simulation. Different mean tilt angles $\tau_0$, i.e., $\tau_0 = 0, \frac{\pi}{6}, \frac{\pi}{4}, \frac{\pi}{3}$ and $\frac{\pi}{2}$ are considered, with $\Delta\tau$ ranging from 0 to $\frac{\pi}{2}$. The particles are perfectly aligned with $\tau = \tau_0$ when $\Delta\tau = 0$; $\Delta\tau = \frac{\pi}{2}$, on the other hand, implies a completely random tilt. The incidence $\theta$ varies from the nadir-looking ($\theta = 0$) to grazing incidence geometry ($\theta = \frac{\pi}{2}$) at every $\frac{\pi}{18}$. Figure 3 presents the simulation results of $Z_{DR}$, $H$ and $\alpha$ on oblate particles, and the results of prolate particles are displayed in Figure 4. The leftmost side of both Figures presents the particle orientation geometry under different mean tilt angles. The simulation results of $Z_{DR}$, $H$ and $\alpha$ are shown in the first, second and third columns, respectively. Different color curves correspond to different incidences; from red to violet, $\theta$ is taken as $0, \frac{\pi}{18}, \frac{\pi}{9}, \frac{\pi}{6}, \frac{2\pi}{9}, \frac{5\pi}{18}, \frac{\pi}{3}, \frac{7\pi}{18}, \frac{4\pi}{9}, \frac{\pi}{2}$. The mean tilt angle $\tau_0$ is 0 (first row), $\frac{\pi}{6}$ (second row), $\frac{\pi}{4}$ (third row), $\frac{\pi}{3}$ (fourth row) and $\frac{\pi}{2}$ (fifth row), respectively.

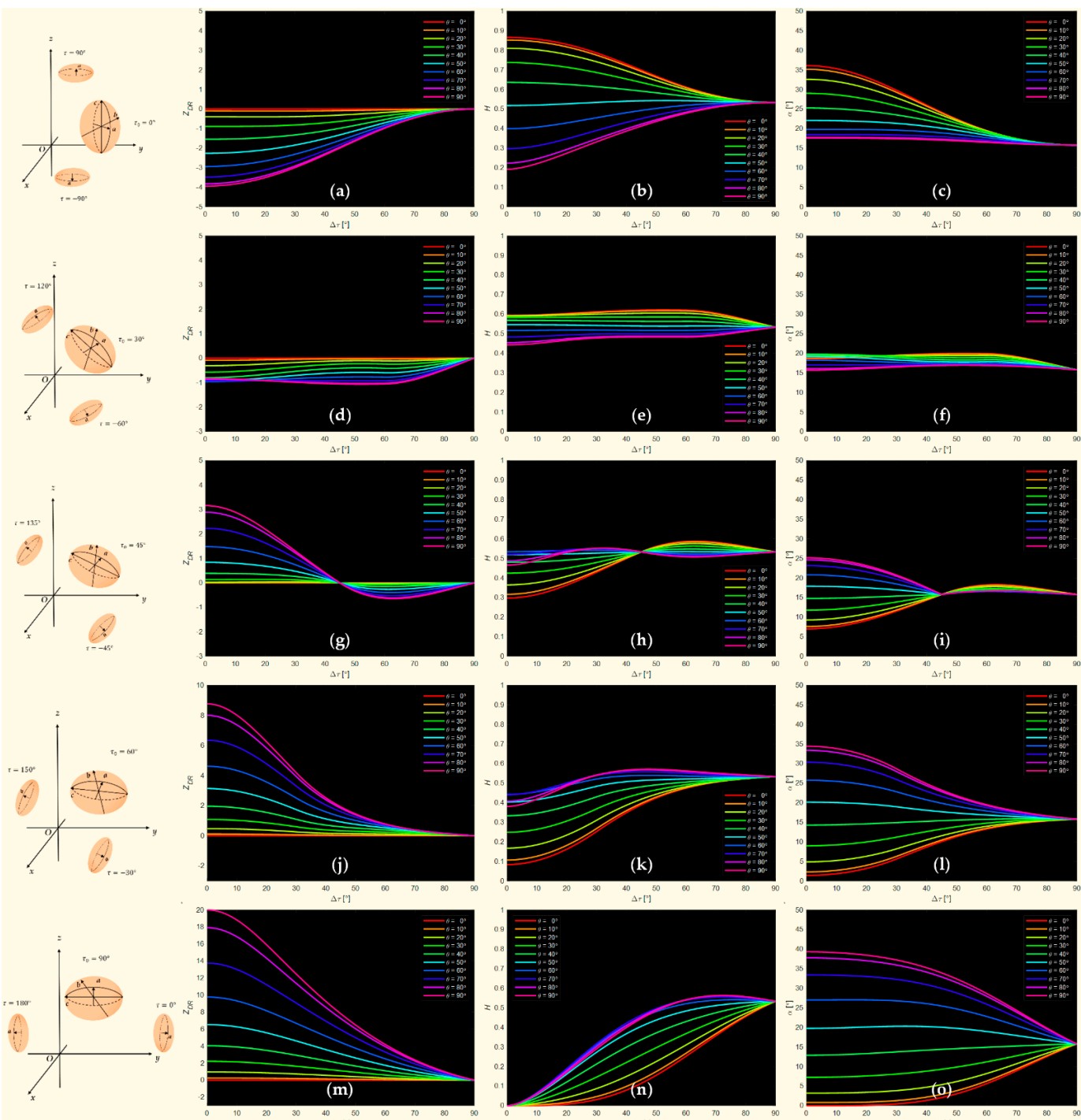

**Figure 3.** Polarimetric descriptors, including (**a,d,g,j,m**) the differential reflectivity $Z_{DR}$, (**b,e,h,k,n**) the polarimetric entropy $H$ and (**c,f,i,l,o**) the mean alpha angle $\alpha$ simulated on a volume of reflection-symmetric (i.e., the canting width $\Delta\varphi = \pi$) oblate particles ($\rho_\Sigma = 11, \rho_\Delta = -9$) with incidence $\theta$ and tilt width $\Delta\tau$ varying from 0 to $\frac{\pi}{2}$, and mean tilt $\tau_0$ taking the values of (**a–c**) 0, (**d–f**) $\frac{\pi}{6}$, (**g–i**) $\frac{\pi}{4}$, (**j–l**) $\frac{\pi}{3}$ and (**m–o**) $\frac{\pi}{2}$, respectively.

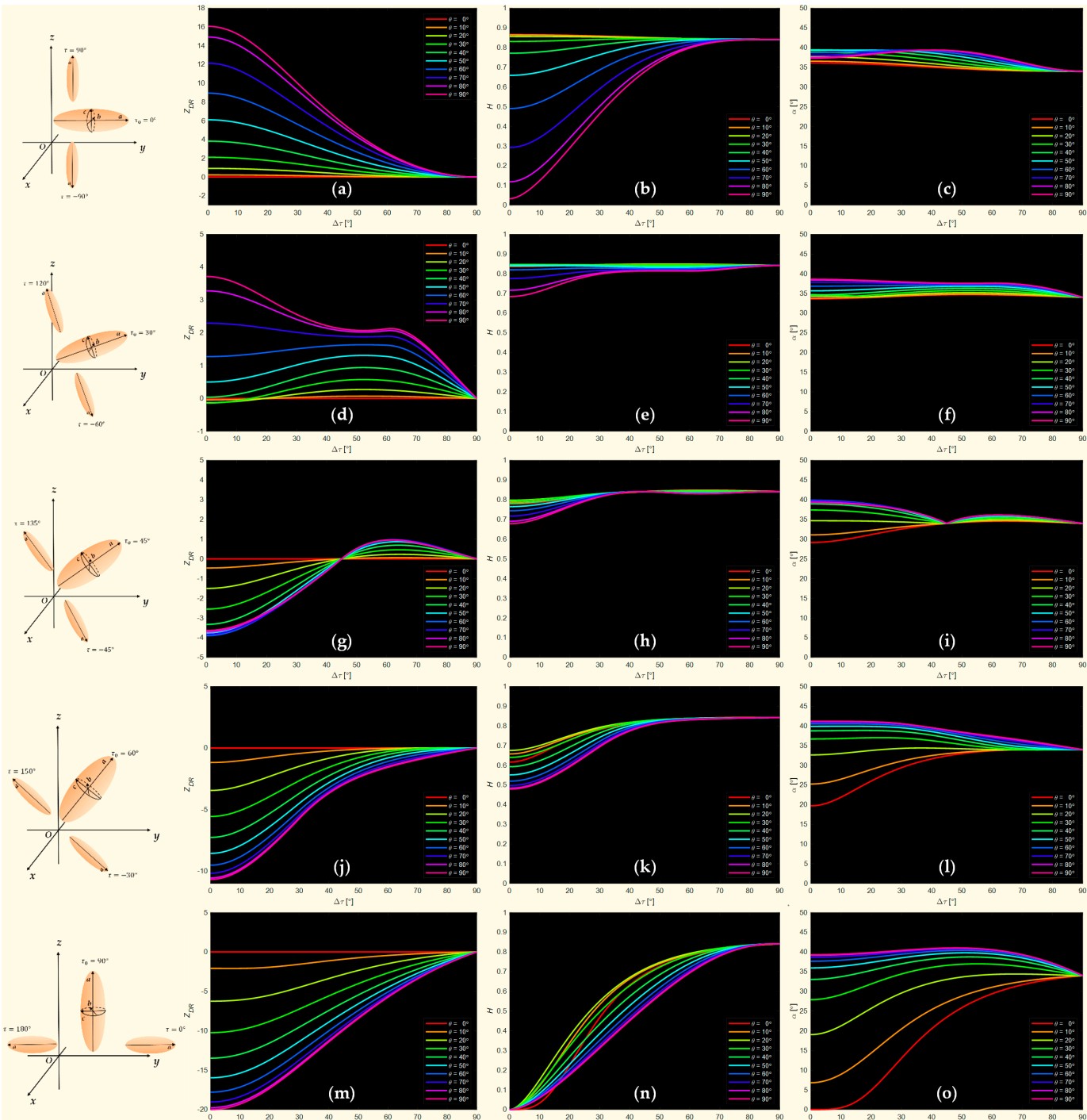

**Figure 4.** Polarimetric descriptors, including (**a,d,g,j,m**) the differential reflectivity $Z_{DR}$, (**b,e,h,k,n**) the polarimetric entropy $H$ and (**c,f,i,l,o**) the mean alpha angle $\alpha$ simulated on a volume of reflection-symmetric (i.e., the canting width $\Delta\varphi = \pi$) prolate particles ($\rho_\Sigma = 11$, $\rho_\Delta = 9$) with incidence $\theta$ and tilt width $\Delta\tau$ ranging from 0 to $\frac{\pi}{2}$, and mean tilt $\tau_0$ taking the values of (**a–c**) 0, (**d–f**) $\frac{\pi}{6}$, (**g–i**) $\frac{\pi}{4}$, (**j–l**) $\frac{\pi}{3}$ and (**m–o**) $\frac{\pi}{2}$, respectively.

Differential reflectivity $Z_{DR}$ is related to the orientation of particles. If the particles are randomly oriented, the reflectivities of horizontal and vertical polarizations will be equal, which leads $Z_{DR}$ to be approximately 0dB, according to Equation (57). $Z_{DR}$ is positive for horizontally tilted particles with $\left\langle |S_{HH}|^2 \right\rangle > \left\langle |S_{VV}|^2 \right\rangle$, and negative for vertically tilted ones with $\left\langle |S_{HH}|^2 \right\rangle < \left\langle |S_{VV}|^2 \right\rangle$. It is shown in Figures 3 and 4 that $Z_{DR}$ varies

significantly with $\theta$. The sign of $Z_{DR}$, nevertheless, is independent of $\theta$. For oblate particles, $Z_{DR}$ is negative and decreases as $\theta$ if $\tau_0 < \frac{\pi}{4}$ or $\{\tau_0 = \frac{\pi}{4}, \Delta\tau > \frac{\pi}{4}\}$. Meanwhile, if $\tau_0 > \frac{\pi}{4}$ or $\{\tau_0 = \frac{\pi}{4}, \Delta\tau < \frac{\pi}{4}\}$, $Z_{DR} > 0$ and increases with $\theta$. The sign of $Z_{DR}$ and the trend of variation with $\theta$ for prolate particles are opposite to those of oblate particles. Hence, the sign of $Z_{DR}$ is affected by the orientation and shape of particles. Moreover, the simulations show that the orientation of particles can be deduced via $Z_{DR}$. When $\tau_0 < \frac{\pi}{4}$, as shown in Figure 3, the oblate particles are biased towards vertically oriented ($Z_{DR} < 0$), and with the increase in $\tau_0$, the particles gradually begin to show a horizontal orientation ($Z_{DR} > 0$). Compared with oblate particles, the orientation of prolates changes in the opposite trend with $\tau_0$, so the sign of $Z_{DR}$ is reversed. Nevertheless, for a fully random tilt orientation ($\Delta\tau = \frac{\pi}{2}$), $Z_{DR}$ converges to 0dB in both cases, as the reflectivities of horizontal and vertical polarizations are equal.

Polarimetric entropy $H$ reflects the degree of confusion of the scattering mechanism. The particle ensemble is weakly depolarized when $H$ is low, and entirely depolarized if $H = 1$. The simulations show that $H$ is significantly influenced by $\theta$, regardless of particle shape, when $\tau_0 = 0$ or $\frac{\pi}{2}$. For oblate particles, the variation of $H$ with $\theta$ is also affected by the particle orientation. If $\tau_0 < \frac{\pi}{4}$ or $\{\tau_0 = \frac{\pi}{4}, \Delta\tau > \frac{\pi}{4}\}$, i.e., particles are vertically oriented, $H$ decreases with the increase in $\theta$; concurrently, $H$ increases with $\theta$ for horizontally tilted particles if $\tau_0 > \frac{\pi}{4}$ or $\{\tau_0 = \frac{\pi}{4}, \Delta\tau < \frac{\pi}{4}\}$. For prolate particles, however, the $H$-$\theta$ variation is unrelated to the orientation, the degree of confusion of scattering mechanism decreases with $\theta$. In both cases, under any $\theta$ and $\tau_0$, as $\Delta\tau$ increases, $H$ converges to 0.5 for oblate particles and to 0.9 for prolate particles. Meanwhile, under the same tilt geometry and incidence, $H$ of prolate particles is always higher than that of oblate particles.

Scattering $\alpha$ angle is related to the physical scattering mechanism. Low $\alpha$ occurs over the smooth regions, which is indicative of dominant surface scattering, while medium and high $\alpha$ occurs over the regions dominated by double-bounce scattering or volume scattering [62]. Based on the simulation results, we find that in both cases, with the increase in $\tau_0$, $\alpha$ is increasingly influenced by $\theta$. Like $H$, for oblate particles, the $\alpha$-$\theta$ variation is also affected by the particle orientation. If particles are vertically oriented, $\alpha$ decreases with the increase in $\theta$, while it increases with $\theta$ for horizontally oriented particles. Meanwhile, for the prolate particles, $\alpha$ increases with $\theta$ regardless of $\tau_0$. As $\Delta\tau$ increases, all curves converge at $\Delta\tau = \frac{\pi}{2}$. Under the same tilt geometry and $\theta$, $\alpha$ of prolate particles is higher than that of oblate particles. Because $\alpha$ is controlled by particle shape, the disk-like oblates have relatively lower $\alpha$ than the tapered prolates.

In summary, the simulations illustrate the excellent performance of the volume scattering model to characterize the polarimetric behaviors of particles with different shapes under different observation and orientation geometries. Thus, we determine that:

(1)  Sign of $Z_{DR}$ is related to the shape and orientation of particles but independent of incidence $\theta$.

(2)  Entropy $H$ is influenced by particle shape and orientation; $H$ of prolates is higher than that of oblates.

(3)  The scattering $\alpha$ angle is also determined by particle shape and orientation; $\alpha$ of prolates is higher than that of oblates. $\alpha$ of prolates with vertical orientation is higher than that of prolates with horizontal orientation. Moreover, $\alpha$ of oblates is more easily affected by the incidence $\theta$. The $\alpha$-$\theta$ variation becomes more significant as the mean tilt angle $\tau_0$ increases.

## 5. Adaptive Polarimetric Decomposition of Volume Scattering Component for Sea Ice

Scattering modeling is dedicated to better describing the underlying components of sea ice scattering. However, the application of the model needs to be associated with polarimetric decompositions to extract parameters used for target identification and classification, such as the scattering powers. In the model-based sea ice decompositions developed by Sharma et al. [40], Zhang et al. [45] and Parrella et al. [46,49], there exists a residual component in the decomposition which can be minimized to the $L_2$ norm but cannot be

eliminated, leading to the incomplete utilization of polarimetric DoFs. Moreover, to facilitate the calculations, the existing sea ice decompositions only consider a model of fixed orientation, such as the random orientation described by Sharma et al. [40] or the vertical orientation detailed by Zhang et al. [45] and Parrella et al. [46]. In fact, vertical, horizontal and random orientations are all possible for particles within sea ice. This requires the decomposition algorithm to adaptively select the appropriate scattering model according to the different orientations.

### 5.1. Scattering Models for Decomposition

To demonstrate the application of the proposed volume models in the decomposition of the backscattering of sea ice, like Zhang et al. [45], we also express the coherency matrix $[T]$ as the incoherent sum of rough surface component $[T_S]$ from brash ice, double-bounce component $[T_D]$ from deformed ice and volume component $[T_V]$ from brine inclusions:

$$[T] = f_S[T_S] + f_D[T_D] + f_V[T_V] \tag{60}$$

where $f_s$, $f_D$ and $f_V$ denote the contributions of the three components. The air-ice interface is generally modeled as a rough surface in view of the fact that the radar return of sea ice surface is significantly influenced by the surface roughness as well as the strong dielectric discontinuities between air and ice [46]. Therefore, if the surface of sea ice is slightly rough (such as brash ice and smooth floe), the surface scattering mechanism will occur. This is characterized by a coherency matrix $[T_S]$ according to the Bragg model [22]:

$$[T_S] = \frac{1}{1+|\beta|^2} \begin{bmatrix} 1 & \beta^* & 0 \\ \beta & |\beta|^2 & 0 \\ 0 & 0 & 0 \end{bmatrix}, \beta = \frac{R_H - R_V}{R_H + R_V} \tag{61}$$

where $R_H$ and $R_V$ are the Bragg coefficients of horizontally and vertically polarized waves. The moving floe ice will collide and converge under the influence of ocean currents, which deforms the surfaces of sea ice into dihedral structures, such as the deformed floe ice and ice ridge. When polarized EMW interacts with these structures, the double-bounce scattering mechanism will occur. This is expressed by a coherency matrix $[T_D]$ according to FDD [22]:

$$[T_D] = \frac{1}{1+|\alpha|^2} \begin{bmatrix} |\alpha|^2 & \alpha^* & 0 \\ \alpha & 1 & 0 \\ 0 & 0 & 0 \end{bmatrix}, \alpha = \frac{e^{j\gamma_H}R_{SH}R_{DH} + e^{j\gamma_V}R_{SV}R_{DV}}{e^{j\gamma_H}R_{SH}R_{DH} - e^{j\gamma_V}R_{SV}R_{DV}} \tag{62}$$

where $R_{SH}$ and $R_{SV}$ denote the h-pol and v-pol Fresnel reflection coefficients of the surface of sea ice; $R_{DH}$ and $R_{DV}$ indicate the corresponding coefficients of the deformed floe ice; $\gamma_H$ and $\gamma_V$ are the respective phases of h-pol and v-pol channels.

To facilitate calculation and analysis, Sharma et al. [40], Nghiem et al. [52] and Zhang et al. [45] simplified the volume scattering model from the aspects of the canting geometry, tilt geometry and shape of brine inclusions within sea ice:

(1) The brine inclusions are canted with no preference [45,52], i.e., $\Delta\varphi = \pi$. This leads to the canting vectors $\boldsymbol{\varphi_s}$ and $\boldsymbol{\varphi_a}$ in Equation (53), and a RS-version of $[T_V]$ is obtained.

(2) The brine inclusions are tilted in the vertical direction, i.e., $|\tau_0| = \frac{\pi}{2}$, $\Delta\tau = 0$ [45,52]. This leads to the tilt vector $\boldsymbol{\tau_{v0}}$ formulated in Equation (36b), and the special Case 5b) in Equation (38) is achieved.

(3) The brine inclusions are approximated by dipoles, as the size of brine inclusions in sea ice is usually at millimeter or even submillimeter level, smaller than the working wavelength of the existing spaceborne PolSAR systems [65–67]. As a result, $\rho_b = \rho_c = 0$, i.e., $\rho_\Sigma = \rho_\Delta = \rho_a$, and $\rho_a$ is typically 1, without loss of generality.

For convenience of decomposition, we adopt the simplifications on canting geometry and shape by characterizing the brine inclusions as a cloud of independent dipoles canted with RS. Nevertheless, considering the reality that the brine inclusions within sea ice may

not always be aligned and uniquely tilted in the vertical direction, we propose CT as a better modeling of tilt geometry. Specifically, we choose the CT geometry shown in Figure 2c, i.e., $\tau_{h0} = \tau_0 = 0$, $|\tau_{v0}| = \frac{\pi}{2}$, and assign $\left[T_V^h\right]$ and $\left[T_V^v\right]$ the same width of tilt ranges, i.e., $\Delta\tau_h = \Delta\tau_v = \Delta\tau = \frac{\pi}{4}$. This provides three volume models of $\left[T_V^h\right]$, $\left[T_V^v\right]$ and $\left[T_V^r\right]$, enabling complete coverage of the horizontal, vertical, and random tilt geometries of brine inclusions, as illustrated in Figure 5. Thus, we obtain the following simplified expressions of $\left[T_V^h\right]$, $\left[T_V^v\right]$ and $\left[T_V^r\right]$:

$$
\left[T_V^h\right] = \begin{bmatrix} T_{11}^h & T_{12}^h & 0 \\ T_{21}^h & T_{22}^h & 0 \\ 0 & 0 & T_{33}^h \end{bmatrix}, \quad \begin{cases} T_{11}^h = -\dfrac{\cos^4\theta}{64} + \dfrac{5\cos^2\theta}{32} + \dfrac{209}{960} \\[2mm] T_{12}^h = T_{21}^H = \left(-\dfrac{\cos^2\theta}{64} + \dfrac{7}{64}\right)\sin^2\theta \\[2mm] T_{22}^h = -\dfrac{\cos^4\theta}{64} + \dfrac{3\cos^2\theta}{32} + \dfrac{97}{960} \\[2mm] T_{33}^h = \dfrac{\cos^2\theta}{16} + \dfrac{7}{60} \end{cases} \tag{63a}
$$

$$
\left[T_V^v\right] = \begin{bmatrix} T_{11}^v & T_{12}^v & 0 \\ T_{21}^v & T_{22}^v & 0 \\ 0 & 0 & T_{33}^v \end{bmatrix}, \quad \begin{cases} T_{11}^v = \left(\sqrt{2}+1\right)\left(\dfrac{\cos^4\theta}{64} - \dfrac{5\cos^2\theta}{32} + \dfrac{47}{960}\right) + \dfrac{4}{15} \\[2mm] T_{12}^v = T_{21}^v = \left(\sqrt{2}+1\right)\left(\dfrac{\cos^2\theta}{64} - \dfrac{7}{64}\right)\sin^2\theta \\[2mm] T_{22}^v = \left(\sqrt{2}+1\right)\left(\dfrac{\cos^4\theta}{64} - \dfrac{\cos^2\theta}{32} + \dfrac{31}{960}\right) + \dfrac{2}{15} \\[2mm] T_{33}^v = -\left(\sqrt{2}+1\right)\left(\dfrac{\cos^2\theta}{16} - \dfrac{1}{60}\right) + \dfrac{2}{15} \end{cases} \tag{63b}
$$

$$
[T_V^r] = \begin{bmatrix} \frac{4}{15} & 0 & 0 \\ 0 & \frac{2}{15} & 0 \\ 0 & 0 & \frac{2}{15} \end{bmatrix} \tag{63c}
$$

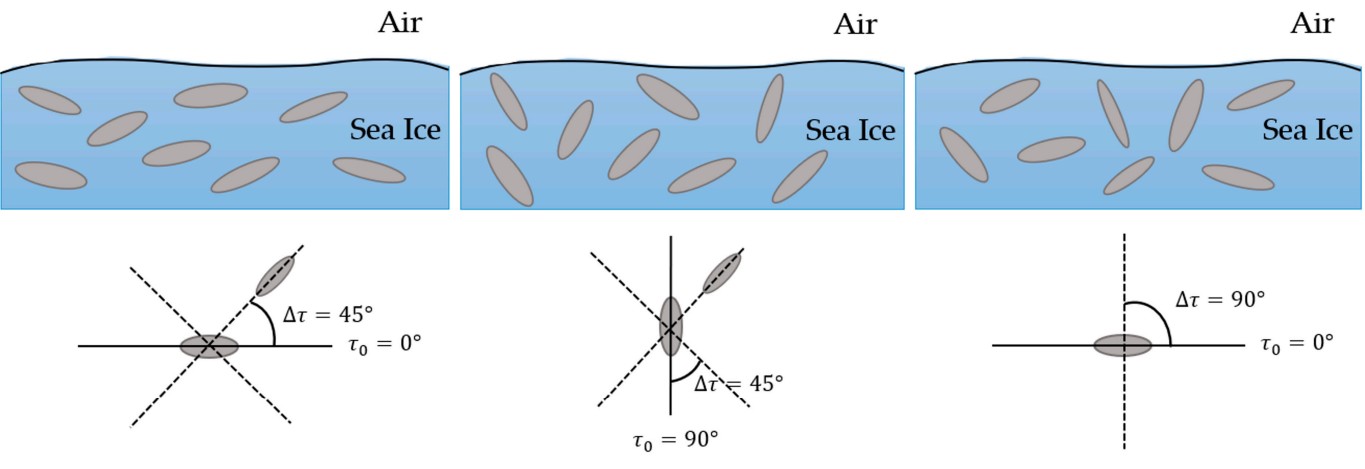

**Figure 5.** Tilt geometries of particles within the sea ice. (**a**) Particles prefer horizontal tilt; (**b**) particles prefer vertical tilt; (**c**) particles show no preference but random tilt.

Hence, the complement relationship in Equation (51) becomes:

$$
\left[T_V^h\right] - [T_V^v] = \sqrt{2}\left([T_V^r] - [T_V^v]\right) \tag{64}
$$

### 5.2. Decomposition Method

From Equations (61)–(63) we can easily determine that:

$$
\mathrm{rank}(f_S[T_S]) = \mathrm{rank}(f_D[T_D]) = 1 < \mathrm{rank}(f_V[T_V]) = \mathrm{rank}([T]) = 3 \tag{65}
$$

where rank($\cdot$) denotes the rank of matrix. For a lossless use of the polarimetric information in $[T]$, it is obvious that $f_V[T_V]$ should reduce rank($[T]$) by exactly one so that:

$$\text{rank}([T_N] \overset{\text{def}}{=} [T] - f_V[T_V]) = 2 \tag{66}$$

This is because we can never extract two rank-1 matrices $f_S[T_S]$ and $f_D[T_D]$ from a full rank-3 matrix $[T_N]$ without any remainder. Hence, there is always a null eigenvector $q$ for $[T_N]$ such that:

$$[T_N]q = 0 \rightarrow [T]q = f_V[T_V]q \rightarrow [T]^{-1}[T_V]q = f_V^{-1}q \tag{67}$$

According to the generalized eigendecomposition in CMD [31], we immediately have:

$$f_V = \frac{1}{\lambda\left([T]^{-1}[T_V]\right)} \tag{68}$$

where $\lambda(\cdot)$ denotes the eigenvalue of a matrix. From the rank reduction point of view, $f_V^{-1}$ can take any of the three positive eigenvalues of the definite matrix $[T]^{-1}[T_V]$. The physical non-negativity of powers $f_S$ and $f_D$, however, requires $f_V$ to be as small as possible so that the remaining rank-2 component $[T_N]$ is also positive semidefinite. As a result, $f_V$ is uniquely determined as:

$$f_V = \frac{1}{\lambda_{\max}\left([T]^{-1}[T_V]\right)} = \lambda_{\min}\left([T_V]^{-1}[T]\right) \tag{69}$$

The volume scattering power is then obtained as follows:

$$P_V = \text{trace}([T_V])f_V \tag{70}$$

where trace($\cdot$) shows the trace of a square matrix. For the three volume scattering models in Equation (63), we can have $P_V^h$, $P_V^v$ and $P_V^r$, respectively, but only one of them is selected according to the adaptive selection strategy. Two selection strategies are provided here.

The first one is based on the maximum power fitting, i.e., to select the maximum of $P_V^h$, $P_V^v$ and $P_V^r$ as the final volume scattering power $P_V$:

$$\begin{aligned} P_V = \max_i(P_V^i) \quad &= \max_i\left(\frac{\text{trace}([T_V^i])}{\max_j\left(\lambda_j\left([T]^{-1}[T_V^i]\right)\right)}\right) \\ &= \max_i\left(\text{trace}([T_V^i])\min_j\lambda_j\left([T_V^i]^{-1}[T]\right)\right) \end{aligned} \quad, i = \{h, v, r\}, j = \{1, 2, 3\} \tag{71}$$

This strategy is possible because it is easy to obtain from the CT relationship Equation (51) as well as the property of singular value that [68]:

$$\left(\frac{\text{trace}([T_V^v])}{P_V^v} - \frac{\text{trace}([T_V^r])}{P_V^r}\right) + \left(\frac{\cos\tau_0\sin\Delta\tau}{1 - \cos\tau_0\sin\Delta\tau}\right)\left(\frac{\text{trace}([T_V^h])}{P_V^h} - \frac{\text{trace}([T_V^r])}{P_V^r}\right) \geq 0 \tag{72}$$

which indicates that either $P_V^h$, $P_V^v$ or $P_V^r$ may be the maximum. Thus, among the volume models $[T_V^h]$, $[T_V^v]$ and $[T_V^r]$, only the one that can fit the maximum power from $[T]$ without violating the physical rank-reduction principle is the best candidate for $[T_V]$:

$$[T_V] = \begin{cases} [T_V^h], P_V = P_V^h \\ [T_V^v], P_V = P_V^v \\ [T_V^r], P_V = P_V^r \end{cases} \tag{73}$$

The second adaptive selection strategy origins from the opposite between $\mathrm{sgn}\left(T_{V12}^h\right)$ and $\mathrm{sgn}\left(T_{V12}^v\right)$ in Equation (55), which is manifested on the dipole-induced volume models in Equation (63) directly as the sign of $T_{V12}^h$, $T_{V12}^v$ and $T_{V12}^r$:

$$\begin{cases} T_{V12}^h > 0 \\ T_{V12}^v < 0 \quad ,\theta > 0 \\ T_{V12}^r = 0 \end{cases} \tag{74a}$$

This is consistent with the tilt geometry corresponding to $\left[T_V^h\right]$, $\left[T_V^v\right]$ and $\left[T_V^r\right]$. According to Equation (57), this is equivalent to the sign of $Z_{DR}\left(\left[T_V^h\right]\right)$, $Z_{DR}\left(\left[T_V^v\right]\right)$ and $Z_{DR}\left(\left[T_V^r\right]\right)$:

$$\begin{cases} Z_{DR}\left(\left[T_V^h\right]\right) > 0 \\ Z_{DR}\left(\left[T_V^v\right]\right) < 0 \quad ,\theta > 0 \\ Z_{DR}\left(\left[T_V^r\right]\right) = 0 \end{cases} \tag{74b}$$

Based on this finding, we then adopt a correspondence principle by decomposing $[T]$ only with a volume scattering model of the same sign of $Z_{DR}$; miscorrespondence is not allowed [69]. In reality, $Z_{DR} = 0$ is difficult to obtain. Hence, to achieve a robust selection of models, we use a short interval around 0, e.g., $Z_{DR} \in \left[-\frac{1}{2}, \frac{1}{2}\right]$, as an alternative to $Z_{DR} = 0$. As a result, to decompose the sea ice coherency matrix $[T]$, the $Z_{DR}$-based selection of volume model $[T_V]$ from the candidates $\left[T_V^h\right]$, $\left[T_V^v\right]$ and $\left[T_V^r\right]$ is carried out as follows:

$$[T_V] = \begin{cases} \left[T_V^h\right], Z_{DR} \in \left(\frac{1}{2}, +\infty\right) \\ \left[T_V^v\right], Z_{DR} \in \left(-\infty, -\frac{1}{2}\right) \\ \left[T_V^r\right], Z_{DR} \in \left[-\frac{1}{2}, \frac{1}{2}\right] \end{cases} \tag{75}$$

where the $Z_{DR}$ in Equation (75) denotes that of $[T]$.

It is noted that the above two strategies are in common use, and similar strategies have been used in the existing decompositions. For instance, the maximum power fitting has been adopted in CMD [31], NNED [28], and the unified Huynen dichotomy of radar target [18], while the $Z_{DR}$-based one has been conducted in four-component scattering power decompositions [23,39].

After extracting $f_V[T_V]$ from $[T]$, a dichotomy can be then conducted to the remaining matrix $[T_N]$ through eigendecomposition or model fitting [31,36] for two rank-1 matrices that correspond to the surface component $f_S[T_S]$ and/or double-bounce component $f_D[T_D]$. Taking model fitting for instance, a dichotomy of $[T_N]$ can be expressed as:

$$[T_N] = [T_{NS}] + [T_{NN}] \tag{76a}$$

$[T_{NS}]$ is achieved by [18]:

$$[T_{NS}] = k_{NS}k_{NS}^H, k_{NS} = \frac{[T_N]q}{\sqrt{q^H[T_N]q}}, q = \begin{bmatrix} 0 \\ 0 \\ 1 \end{bmatrix} \tag{76b}$$

where $q$ is the preferable vector. $[T_{NS}]$ and $[T_{NN}]$ represent the surface component $f_S[T_S]$ or double-bounce component $f_D[T_D]$. Their scattering mechanisms can be identified by the decision rule proposed by An and Xie [36]. Nevertheless, the decomposition results for surface and double-bounce components will not be presented because the application

section of this work concentrates on the volume decomposition of sea ice scattering using the proposed volume models.

*5.3. Transmission Effect*

When using the proposed volume scattering models to achieve polarimetric decomposition of sea ice, we should also consider the effect of transmission and refraction [40,45,46], which is mainly reflected in the following transmission matrix $[Y]$:

$$[Y] = c^2 \begin{bmatrix} 1 & ab & b^2 \\ ab & b^2 & ab^3 \\ b^2 & ab^3 & b^4 \end{bmatrix} \tag{77a}$$

where:

$$a = \frac{\sin\theta}{\sin\theta_r}, b = \frac{1}{\cos(\theta - \theta_r)}, c = \frac{\sin 2\theta \sin 2\theta_r}{\sin^2(\theta + \theta_r)} \tag{77b}$$

At the air-ice interface, the radar signal propagates towards the ice layers with a refractive angle $\theta_r$, which is related to the incidence $\theta$ by Snell's law:

$$\theta_r = \sin^{-1}\left(\frac{\sin\theta}{n_{ice}}\right) \tag{78}$$

where $n_{ice}$ is the refractive index of sea ice. The spaceborne multi-polarimetric microwave remote sensing of sea ice reveals that the refractive indexes of sea ice in the Arctic and the Antarctic range between 1.1 and 1.8 [70].

As a result, for a given coherency matrix $[T]$, its counterpart $[T^t]$ with transmission effect is expressed as [62]:

$$[T^t] = [U_3]\left\{\left([U_3]^{\mathrm{H}}[T][U_3]\right) \circ [Y]\right\}[U_3]^{\mathrm{H}}, [U_3] = \frac{1}{\sqrt{2}}\begin{bmatrix} 1 & 0 & 1 \\ 1 & 0 & -1 \\ 0 & \sqrt{2} & 0 \end{bmatrix} \tag{79}$$

where "$\circ$" is the Hadamard product, i.e., the elementwise product. Combining Equation (77) into Equation (79), we finally obtain the coherency matrix $[T^t]$ with transmission effect as

$$[T^t] = c^2 \begin{bmatrix} \frac{(1+b^2)^2 T_{11} + (1-b^2)^2 T_{22} + 2(1-b^4)\mathrm{Re}(T_{12})}{4} & \frac{(1-b^4)(T_{11}+T_{22}) + 2(1+b^4)\mathrm{Re}(T_{12}) + 4jb^2\mathrm{Im}(T_{12})}{4} & \frac{ab\left((1+b^2)T_{13} + (1-b^2)T_{23}\right)}{2} \\ \frac{(1-b^4)(T_{11}+T_{22}) + 2(1+b^4)\mathrm{Re}(T_{21}) + 4jb^2\mathrm{Im}(T_{21})}{4} & \frac{(1-b^2)^2 T_{11} + (1+b^2)^2 T_{22} + 2(1-b^4)\mathrm{Re}(T_{12})}{4} & \frac{ab\left((1-b^2)T_{13} + (1+b^2)T_{23}\right)}{2} \\ \frac{ab\left((1+b^2)T_{31} + (1-b^2)T_{32}\right)}{2} & \frac{ab\left((1-b^2)T_{31} + (1+b^2)T_{32}\right)}{2} & b^2 T_{33} \end{bmatrix} \tag{80}$$

Like $[T]$, $[T^t]$ is also Hermitian. It is as positive definite as $[T]$ because the transmission matrix $[Y]$ in Equation (77a) can be decomposed via Huynen dichotomy [15,18] as:

$$[Y] = k_1 k_1^{\mathrm{H}} + k_2 k_2^{\mathrm{H}}, k_1 = c\begin{bmatrix} 1 \\ ab \\ b^2 \end{bmatrix}, k_2 = \sqrt{1-a^2}bc\begin{bmatrix} 0 \\ 1 \\ 0 \end{bmatrix} \tag{81}$$

$[Y]$ is thus positive semidefinite, and the property of the Hadamard product [68] ensures the positive definiteness of $[T^t]$. $[T^t]$ becomes $[T]$ with no transmission effect (i.e., $\theta_r = \theta$) when $a = b = c = 1$, and a reflection-symmetric $[T]$ will also lead to a $[T^t]$ with RS.

The Hermite and positive definiteness indicate that $[T^t]$ is also a coherency matrix of physical realizability [61]. Therefore, we can substitute $[T_V^t]$ for $[T_V]$ in Equations (69) and (70) for the volume scattering power in consideration of transmission effect. By replacing volume models $[T_V^h]$, $[T_V^v]$ and $[T_V^r]$ with their transmissive counterparts $[T_V^{th}]$, $[T_V^{tv}]$ and $[T_V^{tr}]$ in Equations (71)–(73), the maximum power fitting is also updated for this situation. For the $Z_{DR}$-based adaptive selection strategy, however, the transmission effect not only

influences the volume models but also the parameter $Z_{DR}$. According to the definition of $Z_{DR}$ on coherency matrix $[T]$, the $Z_{DR}$ for transmissive coherency matrix $[T^t]$, (i.e., $Z_{DR}^t$) can be easily obtained from Equation (80) as:

$$Z_{DR}^t = 10\log_{10}\left(\frac{T_{11}^t + T_{22}^t + 2\mathrm{Re}\left(T_{12}^t\right)}{T_{11}^t + T_{22}^t - 2\mathrm{Re}\left(T_{12}^t\right)}\right) = 10\log_{10}\left(\left(\frac{T_{11} + T_{22} + 2\mathrm{Re}(T_{12})}{T_{11} + T_{22} - 2\mathrm{Re}(T_{12})}\right)\left(\frac{1}{b^4}\right)\right) = Z_{DR} + Z_{DR}^\Delta \quad (82)$$

where $Z_{DR}^\Delta$ denotes a correction term introduced by the transmission effect:

$$Z_{DR}^\Delta = 10\log_{10}\left(\frac{1}{b^4}\right) = 40\log_{10}(\cos(\theta - \theta_r)) \quad (83)$$

As a result, the $Z_{DR}$-based adaptive selection in Equation (75) is updated as:

$$[T_V^t] = \begin{cases} \left[T_V^{th}\right], Z_{DR} \in \left(\frac{1}{2} - Z_{DR}^\Delta, +\infty\right) \\ \left[T_V^{tv}\right], Z_{DR} \in \left(-\infty, -\frac{1}{2} - Z_{DR}^\Delta\right) \\ \left[T_V^{tr}\right], Z_{DR} \in \left[-\frac{1}{2} - Z_{DR}^\Delta, \frac{1}{2} - Z_{DR}^\Delta\right] \end{cases} \quad (84)$$

These strategies and models will be validated in the following experiment on decomposition of GF-3 sea ice data of Antarctica.

## 6. Decomposition Experiment on GF-3 PolSAR Data of Antarctica

### 6.1. Dataset in Prydz Bay

The PolSAR dataset used in this decomposition experiment was acquired by GF-3 in QPSI mode from the northeastern part of Prydz Bay in Antarctica on 29 April 2019. The central geographic coordinate of the dataset is 73°6′E, 68°1′S, adjacent to China's Antarctic Zhongshan Station (76°22′14.28″E, 69°22′24.76″S). Prydz Bay is located in the Indian Ocean, where the sea ice is deeply affected by the winds, upwelling and sea surface temperature, and alternates between the freezing period (from March to September) and melting period (from October to February of the next year). The high latitude makes the change in sea ice in Prydz Bay play an important role in the study of the global climate state and variability.

The acquired GF-3 PolSAR data is a Level-1A single look complex product. Therefore, multilook processing and refined Lee filtering (5 × 5) are implemented to suppress the speckle noise using the ESA software of PolSARpro [62]. Figure 6a displays the obtained Pauli RGB composite image. Figure 6b exhibits the ice atlas chart of Antarctica produced by the National Ice Center (NIC, http://ice.aari.aq/, accessed on 20 July 2022) on 2 May 2019, which shows that the sea ice in the study area is mainly young ice. According to WMO sea ice nomenclature [71], young ice signifies a transition stage between nilas and first-year ice, with thickness ranging from 10 cm to 30 cm. It contains many brine inclusions [52], and the horizontal thin section and vertical micrograph of sea ice indicate that an ellipsoid is an idealized model of a brine inclusion [53]. The young ice in some seasons may be covered by a thin layer of snow. The low salinity of snow generally makes this layer a transparent medium with the dominant surface scattering [72], while the subdominant volume scattering is mainly contributed by the brine inclusions within young ice.

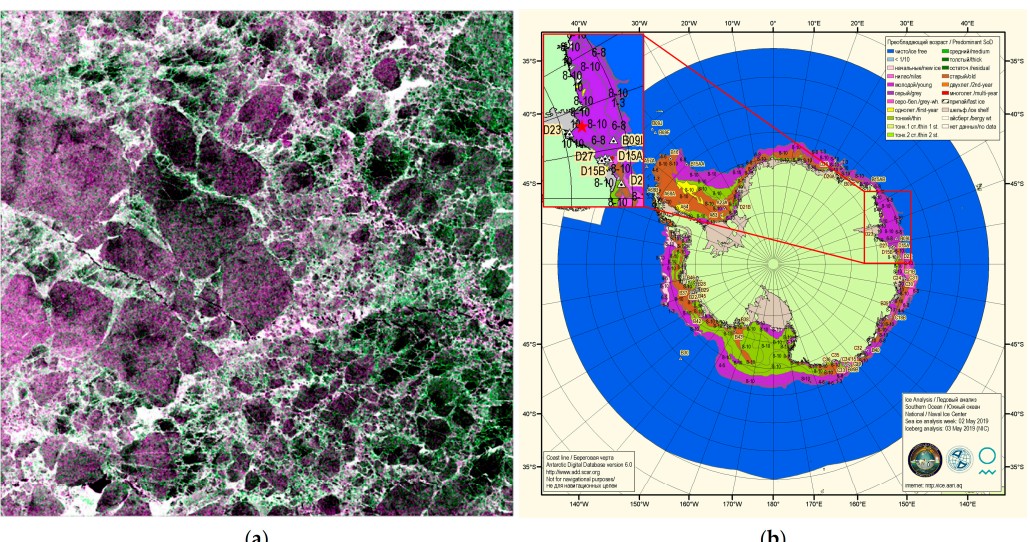

**Figure 6.** Experimental sea ice dataset and information. (**a**) Pauli RGB composite image; (**b**) ice atlas chart of Antarctica produced by the National Ice Center on 2 May 2019. The red star indicates the location of the study area.

### 6.2. Results and Analysis

The PolSAR data were acquired during the Antarctic winter which begins in March and lasts until October. According to [70], the average refractive index is roughly 1.25 (winter) and 1.65 (summer) in the Antarctic. The incidence ranges from the near range 36.91° to the far range 38.45° with the central value $\theta = 37.68°$. An average refractive angle $\theta_r = 29.28°$ is then estimated according to Snell's law. Based on these factors, we obtain the models $\left[T_V^{th}\right]$, $\left[T_V^{tv}\right]$ and $\left[T_V^{tr}\right]$ by bringing the transmission-free volume models $\left[T_V^h\right]$, $\left[T_V^v\right]$ and $\left[T_V^r\right]$ into Equation (80), and adaptively select the $\left[T_V^t\right]$ that matches the coherency matrix $[T]$ at each pixel $(i, j)$ of the image to obtain the power $P_{Vij}$. A power matrix $[P_V] = \{P_{Vij}\}$ is finally produced, where $i = 1, \cdots M, j = 1, \cdots N$, and $M = 1820, N = 1842$ for this dataset. From the perspective of power, the selection among $\left[T_V^{th}\right]$, $\left[T_V^{tv}\right]$ and $\left[T_V^{tr}\right]$ for $\left[T_V^t\right]$ indicates the following three-component decomposition of $[P_V]$:

$$[P_V] = \left[P_V^h\right] + \left[P_V^v\right] + \left[P_V^r\right] \tag{85}$$

where $\left[P_V^k\right]$ $(k = h, v, r)$ is a volume matrix composed by $P_{Vij}^k$, i.e., $\left[P_V^k\right] = \left\{P_{Vij}^k\right\}$. $P_{Vij}^k$ equals to $P_{Vij}$ if model $\left[T_V^{tk}\right]$ is identified as the matched volume model $\left[T_V^t\right]$ for the $(i, j)$ pixel:

$$P_{Vij}^k = \begin{cases} P_{Vij} & , \left[T_V^t\right] = \left[T_V^{tk}\right] \\ 0 & , \text{else} \end{cases}, i = 1, \cdots M, j = 1, \cdots N, k = h, v, r \tag{86}$$

Both the maximum power fitting and $Z_{DR}$-based strategies are adopted. To distinguish between them, we denote the power matrix created by the maximum power fitting as $\left[P_V^{\max}\right]$, which consists of the components $\left[P_V^{\max-h}\right]$, $\left[P_V^{\max-v}\right]$ and $\left[P_V^{\max-r}\right]$; while that achieved by the $Z_{DR}$-based strategy is denoted as $\left[P_V^{\mathrm{zdr}}\right]$, with the components $\left[P_V^{\mathrm{zdr}-h}\right]$, $\left[P_V^{\mathrm{zdr}-v}\right]$ and $\left[P_V^{\mathrm{zdr}-r}\right]$. For this dataset, the correction term $Z_{DR}^{\Delta}$ used in the $Z_{DR}$-based strategy is $-0.1876$, which is estimated by combining the angles $\theta$ and $\theta_r$ into Equation (83).

Figure 7a,b isplay the obtained $\left[P_V^{\max}\right]$ and $\left[P_V^{\mathrm{zdr}}\right]$. Our first impression is of the good consistency between them, as the two strategies achieve the same volume power, i.e., $P_{Vij}^{\max} = P_{Vij}^{\mathrm{zdr}}$, on 64.25% of the image. Nevertheless, for the remaining 35.75%, $P_{Vij}^{\max}$ is

always larger than $P_{Vij}^{\text{zdr}}$, for the maximum power fitting identifies the model that maximizes the volume scattering power, as displayed in Figure 7c in terms of $\left[P_V^{\Delta}\right] \overset{\text{def}}{=} \left[P_V^{\max}\right] - \left[P_V^{\text{zdr}}\right]$. The inconsistency is allocated to the three power components as 9.56%, 14.98% and 11.21%, respectively. To illustrate this further, we also exhibit the horizontal, vertical and random components of $\left[P_V^{\max}\right]$ and $\left[P_V^{\text{zdr}}\right]$ in the second and third rows of Figure 7, respectively. The majority of $\left[P_V^{\max}\right]$ and $\left[P_V^{\text{zdr}}\right]$ are decomposed by both strategies to the vertical components $\left[P_V^{\max-v}\right]$ and $\left[P_V^{\text{zdr}-v}\right]$ illustrated in Figure 7e,h. This is also reflected quantitatively in Figure 8, which shows that the vertically titled model $\left[T_V^{tv}\right]$ is matched to about 70% of the data by the two strategies, i.e., the majority of brine inclusions within the sea ice of the study area are tilted in a vertical direction. This result is consistent with the existing understanding of sea ice geometry. Sea ice beneath a thin transition layer becomes columnar in structure, with the principal axis (i.e., c-axis) parallel to the horizontal plane within a few degrees [52]. Such geometry makes the ice platelets vertical. As a result, the ellipsoidal brine inclusions sandwiched between ice platelets are often preferentially tilted in a vertical direction [52]. Nonetheless, the degree of alignment of c-axis to the horizontal plane is also dependent on the depth. The directions of c-axes of polycrystalline formations within the upper layer of sea ice are actually various [73], but they are gradually aligned horizontally within the layer below, with depth. The mechanism of the transition process of c-axis orientations from random to horizontal is known as the geometric selection [53]. The subdominant random power components $\left[P_V^{\max-r}\right]$ and $\left[P_V^{\text{zdr}-r}\right]$ in Figure 7f,i display the random orientations of c-axes. Besides the depth, the orientation of the c-axis may also be affected by the underlying sea current [74], which can cause the brine inclusions that were originally vertically tilted to become randomly or even horizontally tilted. Figure 7d,g show an example of such horizontally tilted geometry of brine inclusions.

The adaptive polarimetric decompositions of GF-3 data of sea ice in Prydz Bay using two different strategies reveal that the orientations of brine inclusions within sea ice prefer not only the dominant vertical tilt, but also the subordinate random tilt and non-negligible horizontal tilt. Therefore, we propose that the sea ice be characterized using the complementally tilted scattering models instead of a simple vertically tilted one because the former enables a complete coverage of the horizontal, vertical, and random tilt geometries of brine inclusions. Nonetheless, comparing the two power images shown in each column of the second and third rows in Figure 7, we can also see the difference between the two decompositions on the same power component, mainly on the random component and horizontal component. This difference is particularly obvious in Figure 8. The random model and horizontal model account for 19.20% and 10.03% of the area in the decomposition based on maximum power fitting, but the proportion of the two models in $Z_{DR}$-based decomposition becomes 28.78% and 1.13%, respectively. Comparatively, we find the maximum power fitting result in Figure 8a more reasonable, this decomposition selects the candidate model by maximizing the power, which is independent of the specific threshold. In contrast, the $Z_{DR}$-based decomposition relies heavily on the $Z_{DR}$ boundaries determined by experience. For example, if we change the formulated boundary between $\left[T_V^{th}\right]$ and $\left[T_V^{tr}\right]$ in (86) from $Z_{DR} = \frac{1}{2} - Z_{DR}^{\Delta}$ to $Z_{DR} = \frac{1}{3} - Z_{DR}^{\Delta}$, a different decomposition is achieved, with the proportion of horizontal and random models approaching that of the maximum power fitting decomposition.

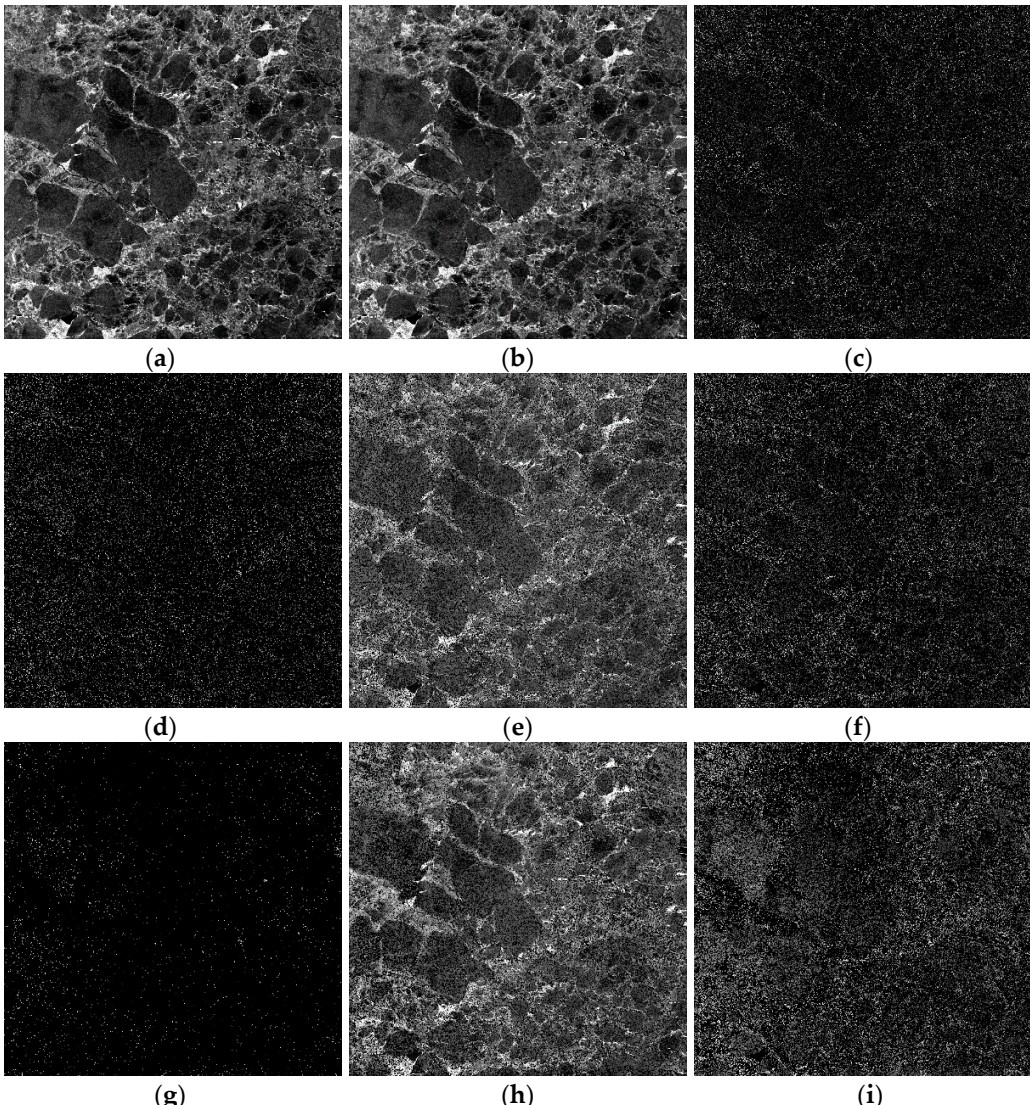

**Figure 7.** Volume powers extracted by (**a**) the maximum power fitting $\left[P_V^{\max}\right]$ and (**b**) $Z_{DR}$-based strategy $\left[P_V^{\mathrm{zdr}}\right]$, as well as (**c**) their difference $\left[P_V^{\Delta}\right]$ and the (**a**,**d**,**g**) horizontal component, (**b**,**e**,**h**) vertical component and (**c**,**f**,**i**) random component of (**d**–**f**) $\left[P_V^{\max}\right]$ and (**g**–**i**) $\left[P_V^{\mathrm{zdr}}\right]$.

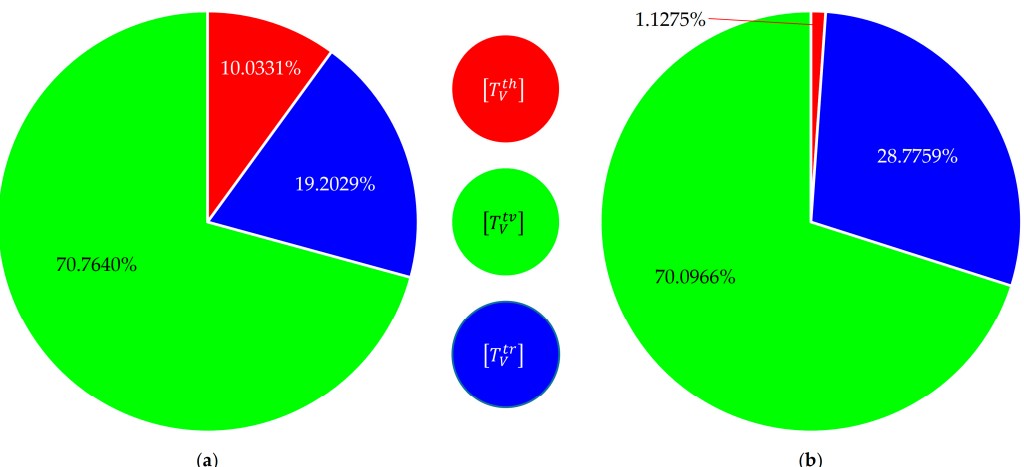

**Figure 8.** Quantitative statistics of the selection of the three tilt scattering models in (**a**) the maximum power fitting and (**b**) $Z_{DR}$-based strategy.

The above analysis reveals that the two adaptive strategies can not only be used in the polarimetric decomposition of sea ice, but can also enable an effective classification of the brine inclusions within sea ice from the viewpoint of tilt geometry. The classification is realized in $Z_{DR}$-based strategy by hard thresholding, as displayed in Figure 9b in terms of the $Z_{DR}$ histogram, where the thresholds $Z_{DR} = -\frac{1}{2} - Z_{DR}^{\Delta}$ and $Z_{DR} = \frac{1}{2} - Z_{DR}^{\Delta}$ classify the ice brine inclusions into the three categories of vertical tilt, random tilt and horizontal tilt, respectively. Such classification, however, is implemented in maximum power fitting via soft thresholding. To illustrate this, we calculate $Z_{DR}$ of each pixel based on Equation (57), and match the achieved $Z_{DR}$ to the three tilt scattering models according to the maximum power fitting result. Then, we create the histogram of $Z_{DR}$ of each model and display them together in Figure 9a in red, green, and blue, respectively. The $Z_{DR}$ histograms resulting from the $Z_{DR}$-based strategy are achieved in the same manner, as illustrated in Figure 9b. Unlike those given in Figure 9b, the histograms of the three categories in Figure 9a overlap one another, showing greater consistency with the general statistical fact. Furthermore, the histograms are also staggered along the $Z_{DR}$ direction in Figure 9a, which clearly illustrates the difference in $Z_{DR}$ among the three models. This distinction is fully driven by the data and model, and does not depend on any empirical thresholds. Here, we are interested in the histogram of the random tilt category. It should distribute around $Z_{DR} = 0$, but is offset in the negative direction of $Z_{DR}$ in Figure 9a. This offset also appears in other histograms of Figure 9. Meanwhile, the randomly tilted model $\left[ T_V^r \right]$ is achieved using Equations (74b) and (82). When the horizontal and vertical polarization channels of radar are well calibrated, the negative offset that occurs in the histogram of the random tilt category is mainly a result of the transmission effect $Z_{DR}^{\Delta}$. As expressed in Equation (83), $Z_{DR}^{\Delta}$ is determined by the incidence $\theta$ and the refractive angle $\theta_r$, and the latter is further related to $\theta$ and the refractive index of sea ice $n_{ice}$ by Snell's law. As a result, given the incidence $\theta$, we may obtain an estimate of $n_{ice}$ using the negative offset $Z_{DR}^{\Delta}$ extracted from the random tilt histogram. As for the GF-3 data of Prydz Bay used in this article, we find that $\hat{Z}_{DR}^{\Delta} = -0.38$ and combine the central incidence $\theta = 37.68°$. The refractive index $n_{ice}$ is finally estimated as 1.4075, which is consistent with the refractive index of Antarctic winter sea ice in previous studies [70,75].

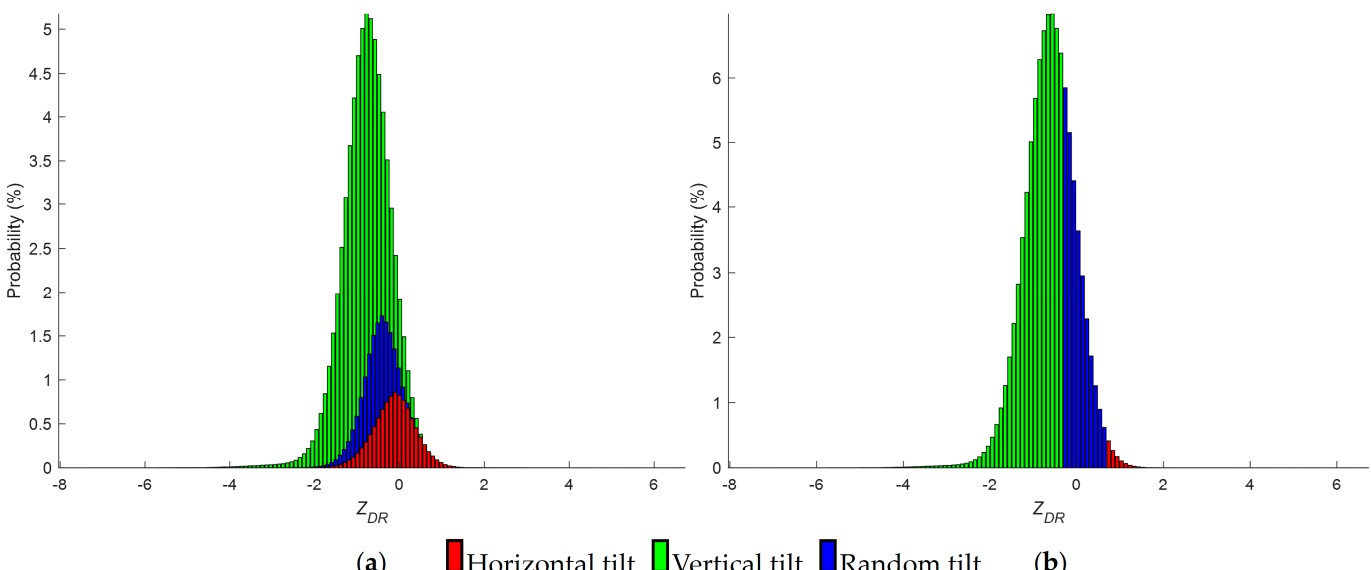

**Figure 9.** $Z_{DR}$ histogram of pixels attributed to the three tilt geometries in (**a**) the maximum power fitting and (**b**) $Z_{DR}$-based strategy.

### 6.3. Discussion

Considering the various orientation distribution of ice particles, we use three volume scattering models to decompose the volume scattering component of sea ice, which can

provide complete coverage of the horizontal tilt, vertical tilt and random tilt of ice particles. We provide two adaptive model selection strategies to adaptively match the volume models according to the orientations of ice particles. One is based on the maximum power fitting and the other is based on $Z_{DR}$. As shown in Figures 7e,h and 8, the majority of the volume scattering powers are decomposed by the two strategies to the vertical components, indicating the brine inclusions within young ice of the study area are preferentially vertically tilted; this is consistent with the previous studies of young ice [52]. Nevertheless, based on the second and third rows of Figure 7 and the quantitative statistics in Figure 8, it is found that when using the two strategies, besides the brine inclusions with vertical tilt, there are also some brine inclusions that show a preference for horizontal random tilt, this is consistent with the geometric selection mechanism, in which the c-axes of polycrystalline formations within sea ice gradually align with depth [53,73]. Therefore, rather than using a simple vertically tilted volume scattering model, it is necessary to adaptively select volume models according to the orientation of ice particles.

The main differences between the two adaptive strategies are the random and horizontal components of volume scattering powers. Based on the quantitative statistics in Figure 8, there is a 9.58% increase in the proportion of the random model in $Z_{DR}$-based decomposition compared with the result in the maximum power fitting-based decomposition, while the proportion of the horizontal model is decreased by 8.9%. This is because the $Z_{DR}$-based strategy relies on the $Z_{DR}$ boundaries, i.e., $Z_{DR} = \frac{1}{2} - Z_{DR}^{\Delta}$, while the maximum power fitting is independent of any specific thresholds and entirely driven by model and data. Thus, we recommend using the maximum power fitting for adaptive model selection. The adaptive decomposition of PolSAR data with the developed volume models may also provide a novel estimation of the refractive index of sea ice. This will be investigated in future work. Moreover, based on the current study, the following two studies on polarimetric radar remote sensing of sea ice will be conducted:

(1)  Take the extinction effects into the developed volume models and decomposition to achieve an estimation of the thickness of sea ice.
(2)  Retrieve the surface and double-bounce scattering components after the volume decomposition to distinguish different types of sea ice.

Although the current work focuses only on the volume modeling and decomposition of sea ice, the developed volume scattering model and the adaptive power decomposition are of general use. The model is built on the 3D uniformly distributed spheroidal particles. It integrates factors such as radar imaging geometry, particle shape, canting geometry, tilt geometry and transmission effects, and it is able to model the polarimetric backscattering of typical distributed volume targets, such as the ice inclusions, vegetation covers and soil particles. The adaptive decomposition can not only enable the model-based interpretation of the polarimetric scattering of sea ice but will also help to realize the successful modeling and classification of complex terrains.

## 7. Conclusions

Model-based polarimetric decompositions have been successfully applied in the interpretation of scattering of sea ice. Volume scattering is an indispensable component of the model-based polarimetric decompositions. The volume scattering of sea ice, especially that of the young ice, is mainly induced by the 3D differently oriented ice brine inclusions. The statistical distribution of particle orientation plays a crucial role in volume scattering modeling of sea ice. Sea ice particles are generally considered to follow the USD. In view of the independence between orientation and distance, USD also implies UOD, i.e., sea ice particles are uniformly oriented in all directions. Nevertheless, we found the existing implementation of UOD was not always effective, while a real UOD could be only realized by distributing the solid angle of particles uniformly in 3D space. We derived the total solid angle of the canting-tilt cell spanned by particles and combined the differential relation between solid angle and Euler angles; thus, the joint PDF $p(\varphi, \tau)$ for UOD was achieved. Starting from the theory of small particle scattering and transformation among ellipsoid

coordinate system, radar polarization coordinate system and Earth-based Cartesian coordinate system, we established the coherency matrix of a $(\varphi, \tau)$-oriented ellipsoid particle. A generalized coherency matrix of a volume of uniformly oriented spheroid particles was then attained by ensembly integrating $p(\varphi, \tau)$ into the $(\varphi, \tau)$-oriented coherency matrix. This covers factors such as radar imaging geometry, particle shape, canting geometry, tilt geometry and transmission effect in a multiplicative way. It can also describe the polarimetric volume scattering of typical distributed targets such as vegetation covers, soil particles and ice inclusions. The existing volume models of sea ice is only applicable to specific cases. The good performance of the model was validated by simulations on a cloud of oblate and prolate particles with differential reflectivity $Z_{DR}$, polarimetric entropy $H$ and scattering $\alpha$ angle as descriptors. To exemplify the generalized model in the adaptive decomposition of the backscattering of sea ice, following CMD, the scattering power that reduces the rank of coherency matrix without breaking the physical realizability was obtained to make full use of the polarimetric information. To adaptively fit the orientation geometry of particles, we studied some interesting orientation geometries, including aligned orientation, complement tilt geometry, reflection symmetry and azimuth symmetry. We highlighted the complement tilt geometry, which provides three models corresponding to the horizontal tilt, vertical tilt and random tilt of sea ice particles, respectively. To match these models to data for adaptive decomposition, we provided two strategies. One is based on $Z_{DR}$, while the other is based on the maximum power fitting. The models and decomposition were finally validated on GF-3 PolSAR data of a young ice area in Prydz Bay, Antarctica. The result demonstrates not only the dominant vertical tilt preference of sea ice brine inclusions, but also the subordinate random tilt preference and non-negligible horizontal tilt preference, which is consistent with the geometric selection mechanism of the c-axis of polycrystalline within sea ice. Compared with the $Z_{DR}$-based strategy, the maximum power fitting is preferable, as it is entirely driven by the model and the data. Such soft thresholding enables it to effectively estimate the $Z_{DR}$ offset introduced by transmission effect, and a novel inversion of sea ice refractive index based on polarimetric model-based decomposition is achieved.

**Author Contributions:** Conceptualization, D.L.; methodology, D.L.; software, D.L. and H.L.; validation, D.L. and H.L.; formal analysis, D.L. and H.L.; investigation, H.L. and D.L.; resources, D.L.; data curation, D.L. and H.L.; writing—original draft preparation, H.L.; writing—review and editing, D.L., H.L. and Y.Z.; visualization, D.L. and H.L.; supervision, D.L. and Y.Z.; project administration, D.L.; funding acquisition, D.L. and Y.Z. All authors have read and agreed to the published version of the manuscript.

**Funding:** This work was supported in part by the National Natural Science Foundation of China under Grants 41871274 and 61971402 and in part by the Strategic High-Tech Innovation Fund of Chinese Academy of Sciences under Grant CXJJ19B10.

**Data Availability Statement:** The GF-3 data are available from the website of China Ocean Satellite Data Service Center (https://osdds.nsoas.org.cn/, accessed on 6 July 2022) after registering and/or ordering.

**Conflicts of Interest:** The authors declare no conflict of interest.

## Appendix A

The matrices $[A]_{mn}$ and $[B]_{mn}$ $(m, n = 1, 2, 3)$ that comprise the essential matrix $[\omega]_{mn}$ for each $(m, n)$ element $T_{Vmn}$ of the $3 \times 3$ volume coherency matrix $[T_V]$ are formulated in this appendix.

$$\begin{cases}[A]_{11} = \begin{bmatrix} \dfrac{\rho_\Delta^2\sin^2\theta\sin 2\theta}{64} & \dfrac{\rho_\Delta^2(\sin^2\theta-4)\sin 2\theta}{64} \\[2mm] \dfrac{3\rho_\Delta^2\sin^2\theta\sin 2\theta}{64} & \dfrac{-\rho_\Delta(13\rho_\Delta\sin^2\theta-4(4\rho_\Sigma+\rho_\Delta))\sin 2\theta}{64} \\[2mm] \dfrac{\rho_\Delta^2\sin^2\theta\sin 2\theta}{32} & \dfrac{-\rho_\Delta(7\rho_\Delta\sin^2\theta-4(2\rho_\Sigma+\rho_\Delta))\sin 2\theta}{32} \end{bmatrix} \\[10mm] [B]_{11} = \begin{bmatrix} \dfrac{\rho_\Delta^2\sin^4\theta}{256} & \dfrac{-\rho_\Delta^2(7\sin^2\theta-6)\sin^2\theta}{64} & \dfrac{\rho_\Delta^2(35\sin^4\theta-40\sin^2\theta+8)}{256} \\[2mm] \dfrac{5\rho_\Delta^2\sin^4\theta}{256} & \dfrac{-\rho_\Delta(11\rho_\Delta\sin^2\theta-2(4\rho_\Sigma+3\rho_\Delta))\sin^2\theta}{64} & \dfrac{\rho_\Delta(15\rho_\Delta\sin^4\theta-24(4\rho_\Sigma-\rho_\Delta)\sin^2\theta+8(8\rho_\Sigma-3\rho_\Delta))}{256} \\[2mm] \dfrac{5\rho_\Delta^2\sin^4\theta}{128} & \dfrac{\rho_\Delta(\rho_\Delta\sin^2\theta+6(2\rho_\Sigma-\rho_\Delta))\sin^2\theta}{32} & \dfrac{-(\rho_\Delta\sin^2\theta+4(2\rho_\Sigma-\rho_\Delta))^2+8(4\rho_\Sigma^2+3(2\rho_\Sigma-\rho_\Delta)^2)}{128} \end{bmatrix}\end{cases} \quad (A1)$$

$$\begin{cases}[A]_{12} = [A]_{21} = \begin{bmatrix} \dfrac{\rho_\Delta^2\cos^2\theta\sin 2\theta}{64} & \dfrac{\rho_\Delta^2(7\sin^2\theta-3)\sin 2\theta}{64} \\[2mm] \dfrac{3\rho_\Delta^2\cos^2\theta\sin 2\theta}{64} & \dfrac{\rho_\Delta(5\rho_\Delta\sin^2\theta-(8\rho_\Sigma+\rho_\Delta))\sin 2\theta}{64} \\[2mm] \dfrac{\rho_\Delta^2\cos^2\theta\sin 2\theta}{32} & \dfrac{\rho_\Delta(\rho_\Delta\cos^2\theta-4\rho_\Sigma)\sin 2\theta}{32} \end{bmatrix} \\[10mm] [B]_{12} = [B]_{21} = \begin{bmatrix} \dfrac{-\rho_\Delta^2(\sin^2\theta-2)\sin^2\theta}{256} & \dfrac{\rho_\Delta^2(25\sin^4\theta-29\sin^2\theta+8)}{256} & \dfrac{-5\rho_\Delta^2(7\sin^2\theta-6)\sin^2\theta}{256} \\[2mm] \dfrac{-5\rho_\Delta^2(\sin^2\theta-2)\sin^2\theta}{256} & \dfrac{\rho_\Delta(29\rho_\Delta\sin^4\theta-(16\rho_\Sigma+33\rho_\Delta)\sin^2\theta+8(4\rho_\Sigma+\rho_\Delta))}{256} & \dfrac{-3\rho_\Delta(5\rho_\Delta\sin^2\theta-2(8\rho_\Sigma+\rho_\Delta))\sin^2\theta}{256} \\[2mm] \dfrac{-5\rho_\Delta^2(\sin^2\theta-2)\sin^2\theta}{128} & \dfrac{-\rho_\Delta(19\rho_\Delta\sin^4\theta+(24\rho_\Sigma-23\rho_\Delta)\sin^2\theta-8(6\rho_\Sigma-\rho_\Delta))}{128} & \dfrac{\rho_\Delta(\rho_\Delta\sin^2\theta+2(4\rho_\Sigma-\rho_\Delta))\sin^2\theta}{128} \end{bmatrix}\end{cases} \quad (A2)$$

$$\begin{cases}[A]_{13} = [A]_{31} = \begin{bmatrix} \dfrac{\rho_\Delta^2(3\sin^2\theta-2)\sin\theta}{64} & \dfrac{-\rho_\Delta^2(5\sin^2\theta-4)\sin\theta}{32} \\[2mm] \dfrac{3\rho_\Delta^2(3\sin^2\theta-2)\sin\theta}{64} & \dfrac{-\rho_\Delta(7\rho_\Delta\sin^2\theta-4(2\rho_\Sigma+\rho_\Delta))\sin\theta}{32} \\[2mm] \dfrac{\rho_\Delta^2(3\sin^2\theta-2)\sin\theta}{32} & \dfrac{-\rho_\Delta(\rho_\Delta\sin^2\theta-4\rho_\Sigma)\sin\theta}{16} \end{bmatrix} \\[10mm] [B]_{13} = [B]_{31} = \begin{bmatrix} \dfrac{\rho_\Delta^2\sin^2\theta\cos\theta}{128} & \dfrac{\rho_\Delta^2(7\cos^2\theta-5)\cos\theta}{64} \\[2mm] \dfrac{5\rho_\Delta^2\sin^2\theta\cos\theta}{128} & \dfrac{\rho_\Delta(11\rho_\Delta\cos^2\theta+(8\rho_\Sigma-9\rho_\Delta))\cos\theta}{64} \\[2mm] \dfrac{5\rho_\Delta^2\sin^2\theta\cos\theta}{64} & \dfrac{-\rho_\Delta(\rho_\Delta\cos^2\theta-(12\rho_\Sigma-\rho_\Delta))\cos\theta}{32} \end{bmatrix}\end{cases} \quad (A3)$$

$$\begin{cases}[A]_{22} = \begin{bmatrix} \dfrac{\rho_\Delta^2(\sin^2\theta-2)\sin 2\theta}{64} & \dfrac{-\rho_\Delta^2(7\sin^2\theta-2)\sin 2\theta}{64} \\[2mm] \dfrac{3\rho_\Delta^2(\sin^2\theta-2)\sin 2\theta}{64} & \dfrac{-\rho_\Delta^2(5\sin^2\theta-6)\sin 2\theta}{64} \\[2mm] \dfrac{\rho_\Delta^2(\sin^2\theta-2)\sin 2\theta}{32} & \dfrac{\rho_\Delta^2(\sin^2\theta+2)\sin 2\theta}{32} \end{bmatrix} \\[10mm] [B]_{22} = \begin{bmatrix} \dfrac{\rho_\Delta^2(\sin^2\theta-2)^2}{256} & \dfrac{-\rho_\Delta^2(7\sin^2\theta-10)\sin^2\theta}{64} & \dfrac{\rho_\Delta^2(35\sin^4\theta-20\sin^2\theta+4)}{256} \\[2mm] \dfrac{5\rho_\Delta^2(\sin^2\theta-2)^2}{256} & \dfrac{-\rho_\Delta^2(11\sin^2\theta-18)\sin^2\theta}{64} & \dfrac{\rho_\Delta^2(15\sin^4\theta-36\sin^2\theta+20)}{256} \\[2mm] \dfrac{5\rho_\Delta^2(\sin^2\theta-2)^2}{128} & \dfrac{\rho_\Delta^2(\sin^2\theta+2)\sin^2\theta}{32} & \dfrac{-\rho_\Delta^2(\sin^4\theta+4\sin^2\theta-20)}{128} \end{bmatrix}\end{cases} \quad (A4)$$

$$\begin{cases}[A]_{23} = [A]_{32} = \begin{bmatrix} \dfrac{\rho_\Delta^2(3\cos^2\theta+1)\sin\theta}{64} & \dfrac{7\rho_\Delta^2\sin^3\theta}{64} \\[2mm] \dfrac{3\rho_\Delta^2(3\cos^2\theta+1)\sin\theta}{64} & \dfrac{5\rho_\Delta^2\sin^3\theta}{64} \\[2mm] \dfrac{\rho_\Delta^2(3\cos^2\theta+1)\sin\theta}{32} & \dfrac{-\rho_\Delta^2\sin^3\theta}{32} \end{bmatrix} \\[10mm] [B]_{23} = [B]_{32} = \begin{bmatrix} \dfrac{\rho_\Delta^2(\cos^2\theta+1)\cos\theta}{128} & \dfrac{7\rho_\Delta^2\sin^2\theta\cos\theta}{64} \\[2mm] \dfrac{5\rho_\Delta^2(\cos^2\theta+1)\cos\theta}{128} & \dfrac{11\rho_\Delta^2\sin^2\theta\cos\theta}{64} \\[2mm] \dfrac{5\rho_\Delta^2(\cos^2\theta+1)\cos\theta}{64} & \dfrac{-\rho_\Delta^2\sin^2\theta\cos\theta}{32} \end{bmatrix}\end{cases} \quad (A5)$$

$$\begin{cases}[A]_{33} = \begin{bmatrix} \dfrac{\rho_\Delta^2\sin 2\theta}{32} & \dfrac{\rho_\Delta^2\sin 2\theta}{32} \\[2mm] \dfrac{3\rho_\Delta^2\sin 2\theta}{32} & \dfrac{3\rho_\Delta^2\sin 2\theta}{32} \\[2mm] \dfrac{\rho_\Delta^2\sin 2\theta}{16} & \dfrac{\rho_\Delta^2\sin 2\theta}{16} \end{bmatrix} \\[10mm] [B]_{33} = \begin{bmatrix} \dfrac{-\rho_\Delta^2\cos^2\theta}{64} & \dfrac{-\rho_\Delta^2\sin^2\theta}{16} & \dfrac{-\rho_\Delta^2(5\sin^2\theta-1)}{64} \\[2mm] \dfrac{-5\rho_\Delta^2\cos^2\theta}{64} & \dfrac{-\rho_\Delta^2\sin^2\theta}{16} & \dfrac{-\rho_\Delta^2(9\sin^2\theta-5)}{64} \\[2mm] \dfrac{-5\rho_\Delta^2\cos^2\theta}{32} & \dfrac{\rho_\Delta^2\sin^2\theta}{8} & \dfrac{-\rho_\Delta^2(\sin^2\theta-5)}{32} \end{bmatrix}\end{cases} \quad (A6)$$

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
