# Peer review of "Solid Angle Geometry-Based Modeling of Volume Scattering with Application in the Adaptive Decomposition of GF-3 Data of Sea Ice in Antarctica"

_remotesensing, doi:10.3390/rs15123208_

Round 1

Reviewer 1 Report

[Overview]

In this paper, the authors developed and demonstrated a model-based PolSAR decomposition process for brine inclusions in sea ice, based on a revised uniform orientation distribution. The technique and the demonstration were well described.

[Comments]

1. Symbols for scalar variables: In the equations, bold symbols were used for both vectors and scalars. This is really confusing. For example, ‘x’ in a bold font was used to express both axis X (vector) and coordinate locations (scalar).

2. In Eq.10, it misses the complex nature of the backscattering coefficients.

3. In general, the two orientation angles, referred to as the canting angle and the tilt angle in this paper, are not independent. In this paper, it seems the canting angle is in fact assumed to be isotropic; this defines a special case in which the assumed independency may stand.

4. In Eq.15 and the text leading to it, an isotropic “canting” angle may again need to be assumed. Otherwise, the two branches may not be equivalent. Likewise, in (23), you may not have the freedom to choose DELTA-Canting-Angle other than Pi.

Reviewer 2 Report

Very appropriate investigation  into model-based polarimetric decomposition of ice. May be further experiments are needed for better confirmation of the proposed model or even on site physical examination of the ice composition.

Author Response

We sincerely thank the Reviewer for the recognition of this work. The performance of the proposed volume models and adaptive decompositions is validated in the current work by simulation and experiments on GF-3 PolSAR data of Antarctica. Nevertheless, on site examination and experiment will help to better confirm the developed models and decompositions further. This is the key work that we need to focus on in the future. We thank the Reviewer for this valuable suggestion.

Reviewer 3 Report

This paper is dedicated to developing a generalized volume scattering model for sea ice particles from the viewpoint of solid angle geometry. From the generalized modeling, three special volume models that satisfy the complementary tilt geometry are obtained for a full coverage of the horizontal tilt, vertical tilt and random tilt of ice particles. Two selection strategies are then proposed to adaptively match the models to particles for the adaptive decomposition. The models and decomposition are validated by simulation and experiments on GF-3 PolSAR data of Antarctica. The result demonstrates the geometric selection mechanism of sea ice from the viewpoint of polarimetric decomposition, and a novel inversion of sea ice refractive index based on polarimetric decomposition is also achieved. Generally, this work is of clear innovation, the manuscript is well-written, and the finding is interesting. I am glad to see the effective application of polarimetric decomposition method in sea ice. Just some minor suggestions are as follows.

1. Introduction. The authors provide a comprehensive overview of volume scattering modeling and the existing polarimetric decomposition methods for sea ice. This is very good. We know that, polarimetric decomposition is an important procedure to extract information regarding sea ice, from which we can further have some other applications, such as classification and inversion of depth of sea ice. Although these are not the focus of the current work, the authors are suggested to add a concise introduction of these decomposition-related applications in their review so as to provide readers with a complete understanding of this work.

2. Modeling. The scattering modeling of a 3D oriented ellipsoidal particle in Equation (8) is interesting. In my impression, Maxim Neumann et al. have conducted the coherent modeling of a spheroid oriented around the LOS of radar. Can Equation (8) cover the case of Neumann et al.’s? Please add some analysis here. Furthermore, what is the meaning of parameter γ in Equation (9)?

3. Decomposition. It is formulated in Equation (60) that the authors in fact consider a model-based polarimetric decomposition with three scattering components, i.e., the surface component Ps*Ts from the rough sea ice surface, the double-bounce component Pd*Td from the deformed floe ice and the volume component Pv*Tv from the ice inclusions. An adaptive polarimetric decomposition of volume scattering component Pv*Tv is developed and detailed in Section 5. Although the decomposition of Ps*Ts and Pd*Td is not the focus of the current work, for completeness, please also briefly introduce the extraction of surface and dihedral components here.

4. Experiment. The two Zdr histograms in Figure 9 are impressive. The histogram on the right is easy to understand because it just comes from the Zdr-based decomposition. However, how do the authors obtain the left histogram for the maximum power fitting-based decomposition? Please add some necessary descriptions here.

5. Writing. Please check the author information in References [1]-[3]. P2L77: “dedicated to pursue” should be “dedicated to pursuing”; P19L550: “dedicated to better describe” should be “dedicated to better describing”.

English Language is good.
